# Doublet microtubule-associated tektins and enzymes differentially regulate sperm flagellar integrity and motility

Qi Liu [1,8], Lunni Zhou[2,3,4,5,8], Xiaochen Liang[2,3,4,5,8], Pengyu Chen[1], Bo Li[1], Shuo Yang[1], Yuqi Liu[1], Haibo Zhao[1], Jin Hu[3,6], Shan Feng [3,6], Shanshan Xie [7], Jianping Wu [2,3,4,5,9] ✉ & Miao Gui [1,9] ✉

Doublet microtubule (DMT)-associated proteins assemble and drive sperm flagella, which are essential for successful fertilization. However, the exact roles of different DMT-associated proteins in sperm function and the underlying molecular mechanisms remain elusive. Here, we generate four gene-knockout mice based on high-resolution structures targeting distinct DMT components: two intermediate filament-like tektins (TEKT1, TEKT5) and two enzymes (TSSK6, DUSP21). The depletion of TEKT1, shared by sperm flagella and motile cilia, causes male infertility characterized by impaired sperm motility and loss of the tektin bundle, whereas sperm-specific *Tekt5* knockout (KO) mice remain fertile with largely normal flagellar function, indicating functional divergence within the tektin family. *Tssk6* KO spermatozoa exhibit severely disturbed morphology and motility, resulting in homozygote infertility and heterozygote subfertility. Phosphoproteomics reveals dysregulated phosphorylation of axonemal proteins, highlighting the critical role of kinase-mediated signaling in regulating sperm motility. Conversely, *Dusp21* KO mice display no fertility or sperm motility defects, suggesting compensatory phosphatase activity. Phenotypic comparisons between *Tekt1* and *Tssk6* KO mice suggest their involvement in distinct subtypes of asthenozoospermia. Overall, this study elucidates how filamentous and enzymatic DMT proteins govern sperm function through divergent mechanisms, which have implications for molecular diagnosis of male infertility.

The sperm flagellum is essential for propelling spermatozoa toward the oocyte during fertilization. Proper assembly and function of the flagellum are crucial for successful reproduction[1,2]. At the center of the flagellum lies the axoneme, a microtubule-based macromolecular machine comprising hundreds of different proteins. The axonemes of

motile cilia and flagella are composed of nine doublet microtubules (DMTs) encircling a central pair of singlet microtubules, generating a conserved "9 + 2" arrangement in diverse cell types, including spermatozoa, respiratory epithelial cells, and unicellular organisms like *Chlamydomonas reinhardtii*[3–5]. Peripheral to the axoneme, sperm

[1]Department of Obstetrics and Gynecology, Sir Run Run Shaw Hospital and Liangzhu Laboratory, Zhejiang University School of Medicine, Hangzhou, Zhejiang, China. [2]State Key Laboratory of Gene Expression, School of Life Sciences, Westlake University, Hangzhou, Zhejiang, China. [3]Zhejiang Key Laboratory of Structural Biology, School of Life Sciences, Westlake University, Hangzhou, Zhejiang, China. [4]Westlake Laboratory of Life Sciences and Biomedicine, Hangzhou, Zhejiang, China. [5]Institute of Biology, Westlake Institute for Advanced Study, Hangzhou, Zhejiang, China. [6]Mass Spectrometry & Metabolomics Core Facility, The Biomedical Research Core Facility, Westlake University, Hangzhou, Zhejiang, China. [7]Children's Hospital, Zhejiang University School of Medicine, National Clinical Research Center for Child Health, Hangzhou, Zhejiang, China. [8]These authors contributed equally: Qi Liu, Lunni Zhou, Xiaochen Liang. [9]These authors jointly supervised this work: Jianping Wu, Miao Gui. ✉e-mail: wujianping@westlake.edu.cn; miaogui@zju.edu.cn

flagella possess sperm-specific structures including outer dense fibers (ODFs) and a mitochondrial sheath in the middle piece, ODFs and a fibrous sheath in the principal piece[6]. Ciliary motility is powered by outer dynein arms (ODAs) and inner dynein arms (IDAs), while regulatory complexes, including radial spokes (RSs) and nexin-dynein regulatory complexes (N-DRCs), orchestrate dynein activity to generate rhythmic waveforms[7]. Unlike cytoplasmic microtubules that associate with a limited number of accessory proteins, ciliary DMTs are densely populated by diverse proteins, including the microtubule inner proteins (MIPs) in their luminal space, which are hypothesized to provide structural stability[8–10]. For instance, in *C. reinhardtii* and *L. mexicana*, mutations in MIPs FAP20, FAP45, or FAP52 result in aberrant ciliary beating and destabilized DMTs, implicating their roles in maintaining axoneme integrity[9,11,12].

High-resolution cryo-electron microscopy (cryo-EM) structures of DMTs isolated from mammalian sperm flagella and epithelial cilia have elucidated their precise protein composition and three-dimensional organization[13–18]. Comparative structural analyses highlight significant interspecies and intercellular diversity in MIP repertoires. For example, sperm DMTs retain nearly all respiratory ciliary MIPs while incorporating approximately a dozen sperm-specific proteins. Among these, tektin proteins constitute a unique family of MIPs that form intermediate filament-like assemblies within the DMT lumen[14]. Both bovine respiratory and sperm DMTs share a pentagonal bundle of parallel tektin filaments composed of TEKT1-4, whereas TEKT5 is located exclusively in sperm and assembles into filaments adjacent to TEKT1-4, suggesting potential functional divergence between conserved and lineage-specific MIPs (Fig. 1). This raises the question of whether shared MIPs like TEKT1 and sperm-specific MIPs like TEKT5 differentially regulate DMT assembly and ciliary motility.

Post-translational modification, including protein phosphorylation, is crucial in regulating sperm motility[19,20]. Testis-specific serine kinases (TSSKs) are expressed in post-meiotic male germ cells and mature mammalian sperm. They are proposed to play indispensable roles in spermatogenesis and sperm function[20]. Intriguingly, recent structural studies have identified TSSK6 and the sperm-specific dual specificity phosphatase 21 (DUSP21) as integral components of mouse sperm DMTs, with both enzymes retaining catalytic activity in vitro[13,17]. However, whether these DMT-associated enzymes primarily play regulatory or structural roles remains to be explored.

Genetic variations in axonemal proteins represent a major etiological factor in male infertility, especially in asthenozoospermia, often disrupting sperm flagellar assembly or motility[21]. How mutations in external axonemal proteins disturb the normal function of spermatozoa is relatively well understood, as their functional impacts are often detectable via traditional transmission electron microscopy (TEM) or immunofluorescence (IF) due to the absence of entire structural modules like ODA. In contrast, the impacts of MIP mutations have been overlooked. In our previous work, cross-referencing the MIP genes with whole-exome sequencing (WES) data of 281 asthenozoospermia patients revealed gene variants in ten distinct MIPs, implicating these genes in flagellar dysfunction[13]. However, definitive validation of their pathogenic potential requires functional interrogation using genetic knockout mouse models, which can clarify genotype-phenotype correlations and mechanistic contributions to sperm motility defects.

To investigate the physiological roles and pathogenic relevance of tektins and sperm DMT-associated enzymes, we generate genetic knockout mouse models for *Tekt1*, *Tekt5*, *Tssk6*, and *Dusp21*. Characterization of the four KO mice shows that *Tekt1* and *Tssk6* KO mice exhibit complete infertility with impaired sperm motility, whereas *Tekt5* and *Dusp21* KO mice retain normal fertility. We explore the detailed molecular mechanisms, highlighting their diverse roles in regulating sperm flagellar architecture and motility. This work positions TEKT1 and TSSK6 as high-priority candidates for molecular diagnostics and therapeutic targeting in human asthenozoospermia.

## Results

### Structures of sperm DMT-anchored tektins and enzymatic proteins

We previously determined the cryo-EM structure of mouse sperm flagellar doublet microtubule (DMT) and built its atomic model[13]. However, some mouse DMT-associated proteins remain unresolved due to limitations in the resolution of cryo-EM densities in certain regions. To address this, we optimized our data processing pipeline, obtaining a 3.0 Å density map of the mouse sperm DMT in the core region (Supplementary Fig. 1, Supplementary Table 1). Guided by the optimized density map and the bovine sperm DMT structure[17], we identified 11 DMT-associated proteins in mouse sperm, including seven sperm-specific proteins (CFAP97D1, EFCAB3, SPMIP1, SPMIP2, SPMIP3, SPMIP4, SPMIP7), and four shared MIPs (SAXO3, CIMIP1, CIMIP3, and CIMIP4), in addition to 47 previously identified DMT-associated proteins[13] (Fig. 1a, Supplementary Fig. 2). Nearly all these mouse sperm DMT-associated proteins were shared by bovine sperm. The only difference between mouse and bovine sperm DMTs is that mouse sperm DMT contains two paralogous MIPs, namely RIBC1 and RIBC2, while bovine sperm omits RIBC1 and employs RIBC2 as a substitute for RIBC1.

A hallmark of mammalian sperm DMT is an intricate organization of tektin filaments within the A-tubule lumen (Fig. 1a). Comparative analysis of tektin architecture across metazoan cilia reveals progressive structural complexity, escalating from invertebrates to vertebrates and from epithelial cilia to sperm flagella (Fig. 1b). Sea urchin sperm DMTs contain three tektin filaments (TEKT1, TEKT2 and TEKT4) adjacent to the ribbon[17]. Human respiratory and oviductal cilia exhibit six tektin filaments arranged in a "V" shape at the cross-section view, whereas other mammalian respiratory and oviductal cilia contain two additional TEKT3 filaments and a linker protein TEKTIP1 occupying the inter-filament groove to form a pentagon arrangement[15,18]. Vertebrate spermatozoa uniquely express TEKT5 that bridges the pentagonal tektin bundle (TEKT1-4) with MIPs along protofilaments A05-A10, resulting in a densely packed MIP architecture in murine and bovine sperm DMTs. Despite this diversity, the filaments of TEKT1, TEKT2, and TEKT4 in the ribbon region are evolutionarily conserved across metazoans, which aligns with the phylogenetic analysis that TEKT3 and TEKT5 emerged later compared to TEKT1, TEKT2, and TEKT4[22] (Fig. 1c). All tektins anchor to the DMT via interactions between TEKT1 and several other MIPs, including FAM166C, SAXO4, EFHC1 and EFHC2 (Fig. 1d). Notably, human *TEKT1* variants were reported to be associated with human asthenozoospermia[23] and loss-of-function study in mice has not been performed before, prompting us to dissect the physiological functions of conserved TEKT1 and sperm-specific TEKT5 using mouse models.

In addition to structural proteins, sperm DMTs harbor several enzymes exhibiting periodic localization. DUSP21, a dual specificity phosphatase anchored to the TEKT3 filament, is present in mouse sperm DMT but absent in human sperm DMT (Fig. 1e,f)[13]. Additionally, DMT possesses an evolutionarily conserved NME7 positioned on the luminal side of the microtubule seam and a sperm-specific kinase TSSK6 localized externally to the seam (Fig. 1e,g).

### Diverse effects of tektins on mouse fertility

To investigate the diverse functions of sperm DMT-associated proteins, we generated four genetic knockout mouse models targeting *Tekt1*, *Tekt5*, *Dusp21* and *Tssk6*, respectively (Supplementary Fig. 3). These models enabled a systematic comparison of the physiological contributions of DMT-associated proteins across multiple functional levels (Fig. 1h).

PCR and sequencing analyses confirmed the absence of the targeted gene sequences in KO mice. Successful gene depletion was further validated by RT-PCR and western blot at the transcription and protein levels, respectively (Supplementary Fig. 3). Fertility

assessments demonstrated that *Tekt5⁻ᐟ⁻* male mice produced litter sizes comparable to WT controls, whereas *Tekt1⁻ᐟ⁻* male mice were infertile (Fig. 2a). The two-cell embryo developmental rate following in vitro fertilization (IVF) remains low in *Tekt1⁻ᐟ⁻* mice, suggesting impaired sperm function under in vitro conditions (Fig. 2b,c). Testicular weight and morphology in both *Tekt5⁻ᐟ⁻* and *Tekt1⁻ᐟ⁻* mice showed no significant differences from WT controls, indicating that testis development is normal (Fig. 2d-f). Hematoxylin-Eosin (HE) staining of the

cauda epididymides and sperm count analyses further confirmed normal spermatogenesis in KO mice (Fig. 3a, Supplementary Fig. 3e), implying that *Tekt1* depletion affects mouse fertility by a distinct mechanism.

To assess potential alterations in sperm morphology and motility, epididymal spermatozoa were capacitated by incubating them in Human Tubal Fluid (HTF) medium supplemented with Fetal Bovine Serum (FBS) for 90 min (Fig. 3b). Fresh *Tekt1⁻ᐟ⁻* spermatozoa exhibited a

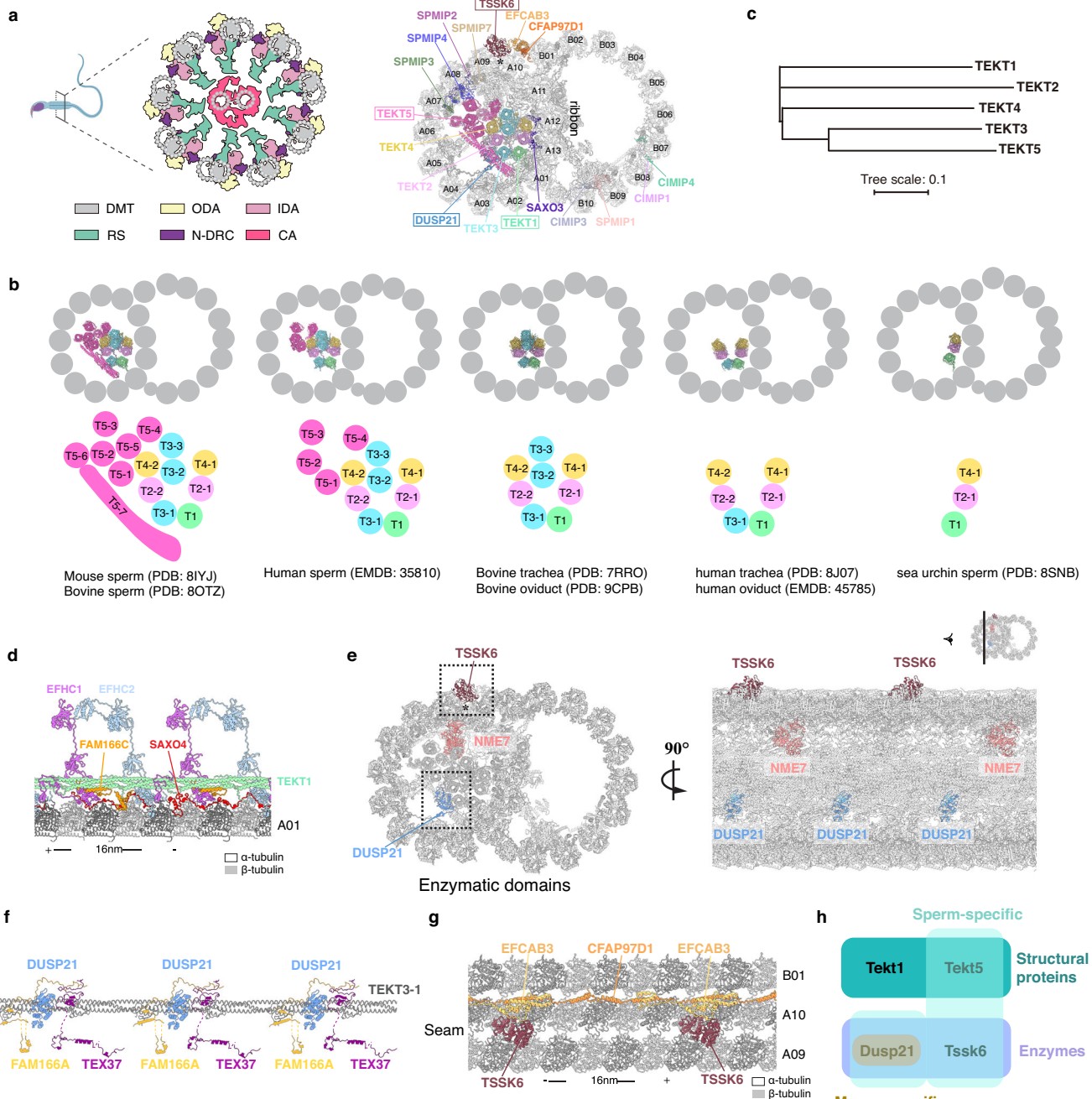

**Fig. 1 | Tektins and enzymatic proteins in mouse sperm DMTs. a** Left: cartoon diagram of mouse sperm axoneme. Right: the atomic model of mouse sperm DMT. Tektin1-5, DUSP21, TSSK6 and 11 newly identified DMT-associated proteins are marked with unique colors. The four proteins studied with mouse models are marked with boxes. The seam of the DMT is marked with an asterisk (*). The sperm diagram was created with BioRender Gui, M. (2026) [https://BioRender.com/jhjkmh2]. **b** Structural comparison of tektin bundles across cell types and species. **c** Phylogenetic tree of mouse TEKT1-5. **d** Structural arrangement of TEKT1 and its

associated MIPs: EFHC1, EFHC2, FAM166C and SAXO4. **e** Localization of enzymatic domain-containing proteins DUSP21, TSSK6 and NME7 in the mouse sperm DMT. **f** Structural arrangement of DUSP21 and its associated MIPs: FAM166A and TEX37. **g** Structural arrangement of TSSK6 and its associated proteins: EFCAB3 and CFAP97D1. **h** The rationality of selecting TEKT1, TEKT5, DUSP21 and TSSK6 for mouse KO study, categorized into sperm-specific, mouse-specific, structural proteins, and enzymes.

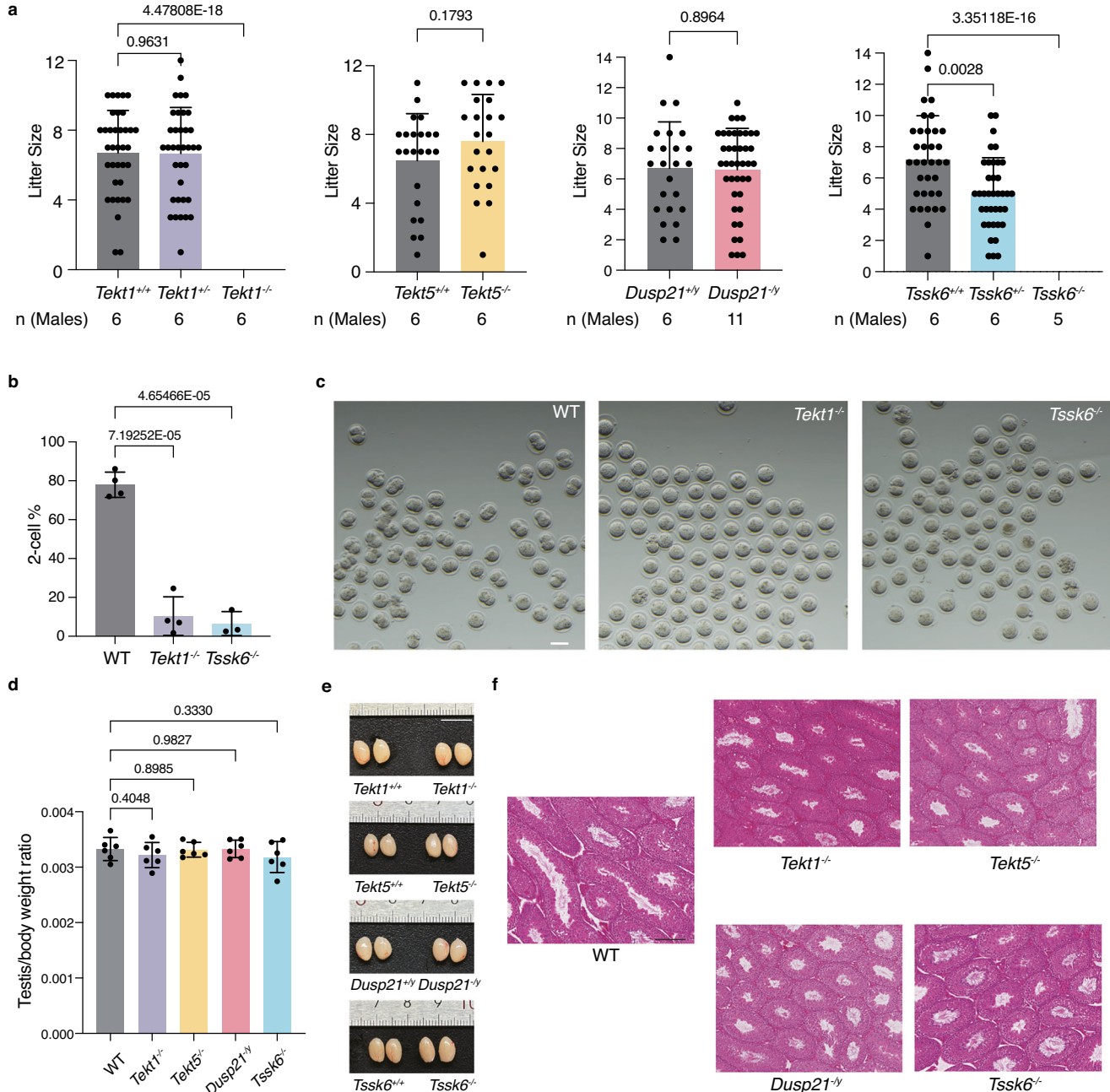

**Fig. 2 | Male fertility analysis of genetic knockout mice. a** Litter size distribution of WT and indicated KO male mice. Each dot represents one litter. Each indicated male mouse was housed together with WT females for three months. The number of male mice per genotype (n) is indicated in the figure. No litter was produced by *Tekt1⁻/⁻* and *Tssk6⁻/⁻* male mice. Statistical significance was assessed using an unpaired two-tailed *t*-test. **b, c** In vitro fertilization rate of WT and indicated mutant spermatozoa (**b**) and corresponding 2-cell images (**c**). n indicates the number of independent male mice used for sperm collection. WT, n = 4; *Tekt1⁻/⁻*, n = 4; Tssk6⁻/⁻,

n = 3. Statistical significance was assessed using an unpaired two-tailed *t*-test. Scale bar: 100 μm. **d** Evaluation of the ratios of testis weight to the body weight of WT and KO mice. n = 6. Statistical significance was assessed using an unpaired two-tailed *t*-test. **e** Images of testes from the indicated mice. Scale bar: 1 cm. **f** Hematoxylin and eosin (HE) stain of testis sections from WT and KO mice. Images shown are representative of three independent biological replicates (individual mice, n = 3). Scale bar: 200 μm. All statistical data are presented as mean ± SD in (**a**), (**b**) and (**d**). The source data underlying Fig. 2 are provided as a Source Data file.

higher proportion (~8%) of coiled morphology, and approximately half displayed bent or coiled flagella after capacitation (Fig. 3b,c, Supplementary Fig. 4). The morphology of the flagella in fresh *Tekt5⁻/⁻* spermatozoa was normal whereas approximately 50% of the spermatozoa transitioned to a bent form after 90 min. Strikingly, computer-assisted sperm analysis (CASA) showed that the motility of *Tekt1⁻/⁻* spermatozoa was severely impaired, consistent with the observed infertility phenotype (Fig. 3e, Supplementary Table 2). In contrast, *Tekt5⁻/⁻* spermatozoa showed a moderate reduction in progressive motility upon

isolation, which became normal after capacitation. High-speed video microscopy revealed a significantly elevated beat frequency in *Tekt1⁻/⁻* spermatozoa compared to WT sperm, and the flagellar waveform showed restricted bending amplitudes (Fig. 3f,g, Supplementary Video 1). Intriguingly, the beat frequencies of both WT and KO spermatozoa tend to decrease after 90 min of capacitation (Fig. 3f). Collectively, these findings demonstrate that *Tekt5⁻/⁻* mice exert minimal effects on fertility and sperm motility, whereas *Tekt1⁻/⁻* mice are infertile with severely impaired sperm motility.

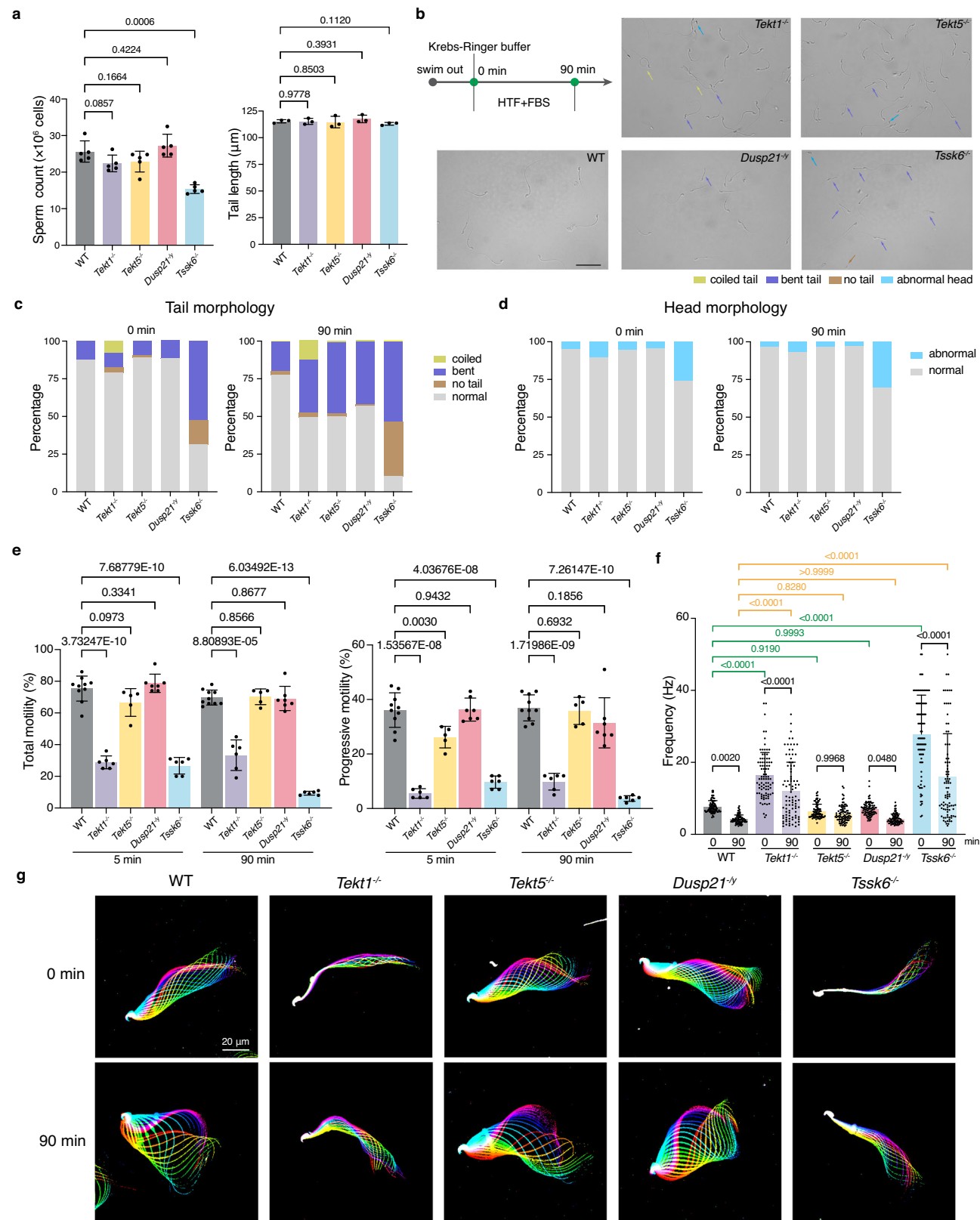

### Structural changes of sperm flagella in *Tekt1* and *Tekt5* KO mice

To analyze the mechanisms underlying the distinct roles of two tektin proteins in sperm motility, we examined the ultrastructure and subcellular localization of axonemal proteins. We randomly checked 30 TEM slices of sperm flagella from each KO mouse. While the *Tekt5*[-/-] sperm exhibited a relatively low rate of "9 + 2" axoneme arrangement abnormality (10%), the *Tekt1*[-/-] sperm showed significantly increased abnormality rate (27%) compared to the WT sperm (7%) (Fig. 4a). In addition, the axonemal disorganization of *Tekt1*[-/-] sperm was observed in both the middle and principal pieces. This observation suggests that the disturbed flagellar axoneme may contribute to the reduced sperm motility of *Tekt1*[-/-] mice.

The axoneme comprises over 200 proteins that assemble into sub-complexes functioning as integrated molecular modules[16,18]. To

**Fig. 3 | Morphology and motility of spermatozoa from knockout male mice.**
**a** Counts and tail length of cauda epididymal spermatozoa. Each dot represents the average value from one mouse. n = 5 for sperm count and n = 3 for tail length. Statistical significance was assessed using an unpaired two-tailed *t*-test.
**b** Representative images of cauda epididymal spermatozoa from indicated mice before capacitation. 0 min indicates non-capacitated spermatozoa freshly collected in Krebs-Ringer buffer; 90 min indicates capacitated spermatozoa incubated in HTF + FBS buffer for 90 min. Representative abnormal spermatozoa were indicated by color-coded arrows. Scale bar: 50 μm. **c**, **d** The statistics of sperm tail (**c**) and head (**d**) morphology before (0 min, left) and after (90 min, right) capacitation. Each column was the mean of at least three biological replications. **e** Motile cells (left) and progressive cells (right) were calculated before (5 min, left) and after

(90 min, right) capacitation, respectively. WT, n = 10; *Tekt1*⁻/⁻, n = 6; *Tekt5*⁻/⁻, n = 5; *Dusp21*⁻/y, n = 7; *Tssk6*⁻/⁻, n = 6. Statistical significance was assessed using an unpaired two-tailed *t*-test. **f** Beat frequency of cauda epididymal spermatozoa before (0 min) and after (90 min) capacitation. Frequency: the number of beat cycles per second. n = 90 in each group (30 spermatozoa from each male, 3 males). Statistical significance was assessed using one-way ANOVA. **g** The flagellar waveform of cauda epididymal spermatozoa before (0 min, upper) and after (90 min, lower) capacitation. Movies were recorded at 200 fps from the head-tethered spermatozoa in fibronectin-coated dishes. Overlays of flagellar traces from a complete beat cycle were colored in a rainbow spectrum, according to chronological order. All statistical data are presented as mean ± SD in panels a, e and f. The source data underlying Fig. 3 are provided as a Source Data file.

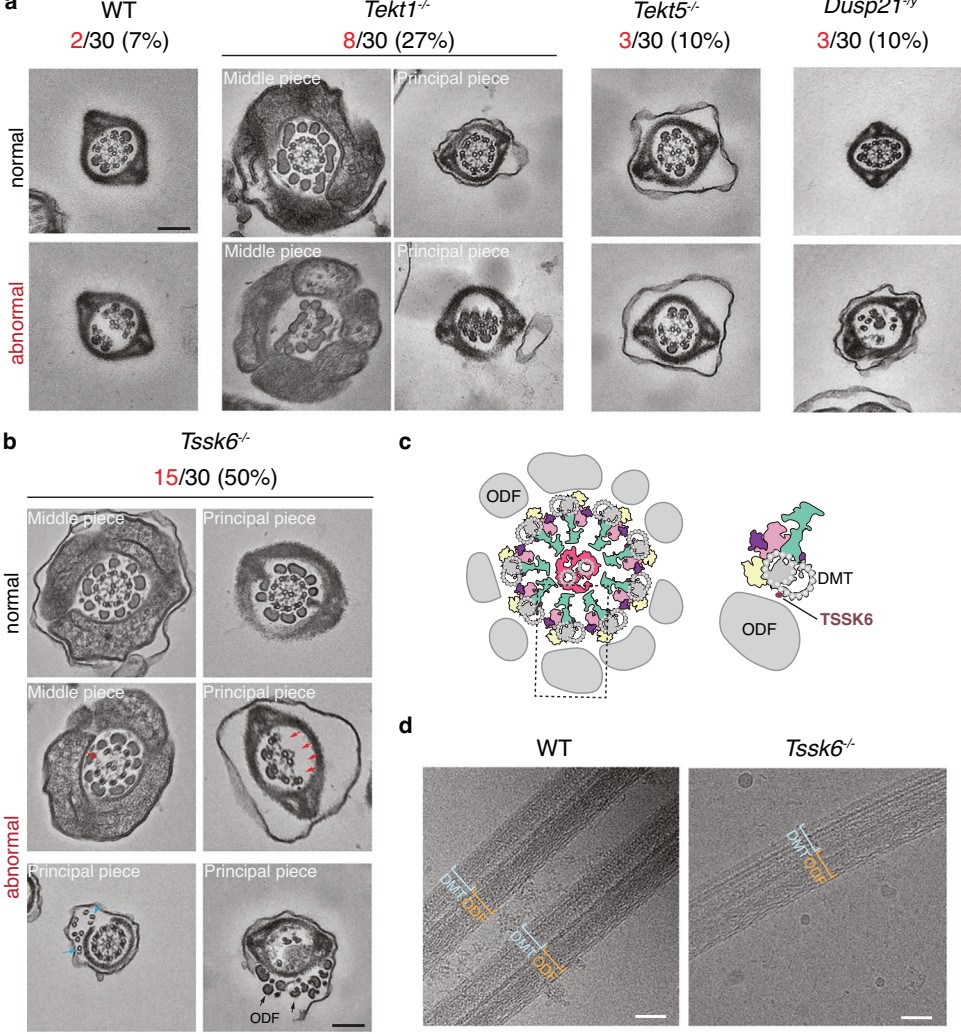

**Fig. 4 | Transmission electron microscopy (TEM) examinations of sperm flagella. a** Representative cross-sectional TEM images of cauda epididymal spermatozoa from WT, *Tekt1*⁻/⁻, *Tekt5*⁻/⁻, *Dusp21*⁻/y mice, showing the normal (top) and abnormal axonemes (bottom). 30 randomly selected cross sections from three mice per genotype were analyzed, and the proportion of abnormal axonemes (abnormal/30) is indicated. Scale bar: 200 nm. **b** Representative cross-sectional TEM images of cauda epididymal spermatozoa from *Tssk6*⁻/⁻ mice, showing the normal (top) and abnormal axonemes (bottom). 30 randomly selected cross sections from three mice per genotype were analyzed, and the proportion of abnormal

axonemes (abnormal/30) is indicated. Note that the abnormalities happen both in the principal and middle pieces. Red arrows: loss of DMT; black arrows: mislocalized ODF and DMT; blue arrow: extra MT. Scale bar: 200 nm. **c** Schematic diagram showing the position of ODF and axoneme. TSSK6 localizes at the junction of ODF and DMT. **d** Representative cryo-EM images showing the reservation of ODFs anchoring on DMTs in WT (n = 15) and *Tssk6*⁻/⁻ (n = 18) spermatozoa. The other images with similar phenotype are provided in a Source Data file. Scale bar: 25 nm. The source data underlying Fig. 4 are provided as a Source Data file.

determine if tektin depletion within the microtubule lumen affects the assembly of other axonemal proteins, we employed immunofluorescence (IF) to evaluate the subcellular localization of core axonemal proteins: CFAP21 (MIP), DNAH1 (IDA), DNAH17 (ODA), RSPH9

(RS) and SPAG6 (N-DRC). Intriguingly, IF staining results of *Tekt1*⁻/⁻ and *Tekt5*⁻/⁻ spermatozoa were indistinguishable from WT controls (Supplementary Fig. 4e), indicating that the spatial organization of these axonemal complexes remains largely preserved despite tektin loss.

Given the limited resolution of conventional TEM and fluorescence microscopy, we determined near-atomic resolution structures of sperm DMTs from KO mice using cryo-EM. Following established protocol for WT sperm DMT sample preparation, flagellar membranes were removed using detergent, and axonemes were splayed into DMTs with ATP (Fig. 5a). Cryo-EM analyses generated a 3.9 Å density map of the 48-nm repeat DMTs from *Tekt5⁻/⁻* mice and a 4.8 Å density map from *Tekt1⁻/⁻* mice (Fig. 5, Supplementary Fig. 5). In *Tekt1⁻/⁻* DMTs, structural perturbations were observed within the A-tubule lumen. Notably, the tektin filaments, not just TEKT1, were almost completely absent, as well as protein densities directly associated with tektins: TEKTIP1, DUSP21, SAXO3, SAXO4, a part of FAM166A, and the C-termini of SPMIP6, TEX37, and FAM166C (Fig. 5d,h, Supplementary Table 3). Other MIPs, such as CFAP21, remained structurally normal, consistent with the IF staining result (Supplementary Fig. 4e). In *Tekt5⁻/⁻* DMTs, TEKT5 and its interacting partners – DUSP21, the C-terminus of TEX37, the N-terminal tail of TEKT3, and fragments of FAM166A – were barely observed (Fig. 5e,i). Intriguingly, in both KO models, the N-terminus of TEX37 retained visible and maintained direct interactions with the microtubule luminal surface (Fig. 5d,e), indicating that TEX37 localization on sperm DMTs persists despite the loss of its C-terminal anchoring partners. Notably, the observed structural loss could result from altered protein expression or mislocalization, which warrants further validation through proteomic analysis.

## Tektin depletions cause changes in the proteomic level

To determine whether tektin depletion alters global protein levels and cellular pathways, we conducted quantitative mass spectrometry (MS) on whole spermatozoa. Compared to WT controls, *Tekt1⁻/⁻* spermatozoa exhibited significant reductions (fold change ≥ 2, *p*-value < 0.05) in 62 proteins and increases in 31 proteins. Gene Ontology (GO) and Kyoto Encyclopedia of Genes and Genome (KEGG) analyses revealed that the down-regulated proteins were enriched in pathways related to flagellar movement and assembly (Fig. 6). Notably, proteins down-regulated in *Tekt1⁻/⁻* spermatozoa, including TEKT1-5, TEKTIP1, DUSP21, SAXO4, and CIMIP2C (FAM166C), were also perturbed in the cryo-EM structure (Fig. 5). In addition, a subgroup of axonemal proteins localized to the axonemal dyneins (ODAD1, DNAH1, DNAI2) and the central apparatus (SPEF2, SPAG17, KIF9, CFAP74, BTBD16, CFAP99) were modestly down-regulated, suggesting moderate global perturbation of the axonemal proteome following TEKT1 loss. Despite these changes, structural analyses showed that the external axonemal complexes, including the N-DRCs, RSs, IDAs, and ODAs, remained largely intact in the 96-nm repeat density map of *Tekt1⁻/⁻* sperm DMTs (Supplementary Fig. 5g), consistent with the IF results (Supplementary Fig. 4e). Notably, near-intact external axonemal structures do not necessarily mean that all structures remain unchanged, as our cryo-EM image processing excluded some poor-quality particles. Taken together, these findings indicate that TEKT1 depletion eliminates the tektin bundle and associated MIPs, which reduces flagellar stability and motility with elevated abnormal morphology, leading to male infertility. In contrast, *Tekt5⁻/⁻* spermatozoa exhibited down-regulation of limited axonemal proteins, including TEKT5 and DUSP21 (Fig. 6d), and the differentially expressed proteins were mapped in pathways unrelated to sperm motility. These results suggest that TEKT5 loss has minimal impact on the assembly of other axonemal components. Together, these findings revise the previous argument that the tektin filament may function as a molecular ruler to guide the docking of axonemal complexes and other MIPs[24].

## Tektins regulate DMT stability

To investigate how *Tekt1⁻/⁻* induced DMT structural defects impair sperm motility, we evaluated the mechanical stability of sperm DMTs. Ciliary DMTs are stabilized by an interconnected network of MIPs, which confer structural resilience to withstand bending forces during ciliary beating[9]. We hypothesized that tektin depletion results in fragile sperm DMTs. Taking advantage of the fact that sonication could disrupt DMT integrity and depolymerize tubulins[25], we quantified free tubulin levels via western blot using an acetyl-tubulin antibody across varying sonication durations (Fig. 7). Compared to WT, *Tekt1⁻/⁻* spermatozoa exhibited accelerated accumulation of free tubulin, suggesting heightened susceptibility to sonication-induced damage. Interestingly, *Tekt5⁻/⁻* spermatozoa released free tubulin at a comparable level to that of *Tekt1⁻/⁻* spermatozoa following ultrasonic treatment. Taken together, these results suggest that loss of tektins renders the DMT mechanically fragile, underscoring their critical role in maintaining DMT structural robustness.

## Characterization of fertility and sperm motility in *Dusp21* and *Tssk6* KO mice

To investigate the physiological roles of DMT-associated enzymes on sperm motility regulation, we analyzed fertility and spermatozoa phenotypes in *Dusp21⁻/y* and *Tssk6⁻/⁻* mice. *Tssk6⁻/⁻* male mice are completely infertile and *Tssk6⁺/⁻* male mice are subfertile (Fig. 2a). Moreover, the percentage of two-cell embryos after IVF was significantly decreased compared with WT (Fig. 2b,c). Both testicular and cauda epididymal morphologies of *Tssk6⁻/⁻* mice appeared normal (Fig. 2d-f, Supplementary Fig. 3e), yet swim-out assays revealed reduced sperm count (Fig. 3a), potentially attributed to motility defects. *Tssk6⁻/⁻* spermatozoa displayed severe morphological abnormalities, including no tail, bent tail, and aberrant sperm head morphology (Fig. 3b-d, Supplementary Fig. 4). Intriguingly, the flagella length was comparable to WT for *Tssk6⁻/⁻* spermatozoa that retained tails (Fig. 3a). Sperm motility was profoundly disrupted, with reduced progressive movement, abnormal beat frequency, and irregular flagellar waveforms (Fig. 3e-g), consistent with previous reports showing that *Tssk6⁻/⁻* mice are male sterile due to marked defects in sperm motility and morphology[26,27] and that polymorphisms in TSSK6 are associated with human male infertility[28]. Notably, sperm tail loss was a distinctive feature of *Tssk6⁻/⁻* spermatozoa (Fig. 3c), suggesting its essential role in connecting the sperm head and tail. This phenotype correlates with disrupted phosphorylation of SPATA6, a key component of the head–tail coupling apparatus[29], in *Tssk6⁻/⁻* spermatozoa (See below and Source Data). The combination of a weakened head–tail junction and elevated beat frequency (Fig. 3f) likely contributes to the progressive tail detachment observed after capacitation (Fig. 3c). In contrast, *Dusp21⁻/y* spermatozoa exhibited normal morphology, motility, and fertility, mirroring WT phenotypes (Figs. 2, 3 and Supplementary Figs. 3, 4).

Cross-sectional TEM showed that 10% of slices display abnormal axoneme in *Dusp21⁻/y* spermatozoa, similar to WT (Fig. 4a). In contrast, half of *Tssk6⁻/⁻* spermatozoa exhibited disorganized axonemal ultrastructure, affecting both the middle and principal pieces (Fig. 4b). Intriguingly, diverse axoneme abnormalities in *Tssk6⁻/⁻* spermatozoa were observed, including loss of one or multiple DMTs, mislocalized DMTs and ODFs, or redundant MTs. TSSK6 localizes to the junction between the DMT and ODF and likely plays a role in mediating structural cohesion between these components (Fig. 4c). Surprisingly, ODFs were correctly positioned at the DMT periphery in *Tssk6⁻/⁻* spermatozoa, and some ODFs even remained attached to mislocalized DMTs, both within and outside the fibrous sheath, suggesting that TSSK6 is dispensable for ODF-DMT linkage (Fig. 4b,d).

In *Dusp21⁻/y* spermatozoa, cryo-EM analysis revealed largely diminished density of DUSP21 and its interacting partner, the C-terminus of TEX37, while other MIPs remained intact (Fig. 5f,j). *Tssk6⁻/⁻* spermatozoa lacked TSSK6 and its associated proteins CFAP97D1 and EFCAB3, while all MIPs were retained (Fig. 5g,k), regardless of the strong phenotype. CFAP97D1 KO male mice exhibit subfertility and sperm motility defects, albeit milder than *Tssk6⁻/⁻* phenotypes[30], underscoring the functional indispensability of the TSSK6-CFAP97D1-EFCAB3 module (Fig. 1g).

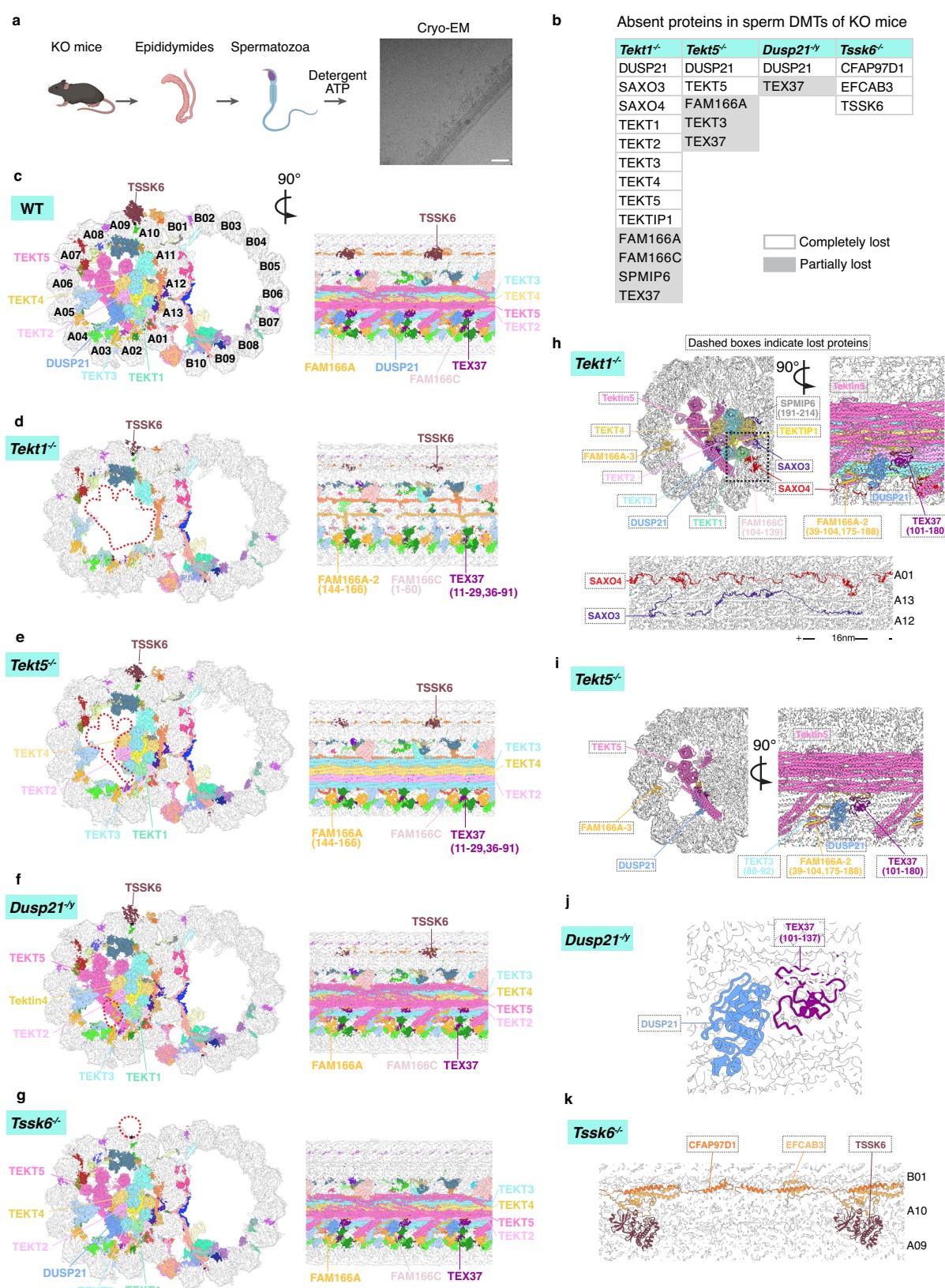

Taken together, these findings demonstrate that *Dusp21⁻/ʸ* mice exhibit largely normal sperm function, whereas *Tssk6⁻/⁻* mice display severe flagellar defects including bent or absent flagella, disrupted motility, and complete male infertility. While ultrastructural analysis revealed disorganized "9 + 2" microtubule arrangements in a subset of *Tssk6⁻/⁻* spermatozoa, cryo-EM reconstructions and IF staining of remaining DMTs showed preserved architecture of MIPs and axonemal complexes (Supplementary Figs. 4e and 5g). This paradox prompts us to ask whether TSSK6 regulates axonemal function not only through affecting structural stability, but also via modulation of sperm protein phosphorylation status that is critical for motility.

**Fig. 5 | Cryo-EM structures of sperm DMTs from knockout mice revealed the absence of selected proteins. a** Isolation of sperm DMTs from KO mice for cryo-EM study. One representative micrograph from 16,707 micrographs of the *Dusp21⁻/y* dataset is shown. Scale bar: 50 nm. Schematic diagram was created with BioRender Gui, M. (2026) [https://BioRender.com/jhjkmh2]. **b** The table summarizes the proteins with absent or discontinuous densities in sperm DMTs of KO mice. Detailed information is available in Supplementary Table 3. **c–g** The cross-sections (left) and longitudinal sections (right) showing the sperm DMT maps from WT, *Tekt1⁻/⁻*, *Tekt5⁻/⁻*, *Dusp21⁻/y* and *Tssk6⁻/⁻* mice colored by subunits. The absent densities of the knockout DMTs are marked with red dashed circles in the cross-sections. The remaining residues of the partially lost proteins are indicated. **h–k** Overlays of the WT DMT model with the density maps of KO DMTs. The missing or discontinued densities are marked with dashed boxes. For the partially lost proteins, missing residues compared with the WT models are indicated.

## Quantitative proteomics and phosphoproteomics of *Dusp21* and *Tssk6* KO mice

To systematically analyze proteomic and phosphoproteomic alterations of *Tssk6⁻/⁻* and *Dusp21⁻/y* spermatozoa, we performed quantitative MS. While a portion of flagella-associated proteins exhibited modest abundance changes in the proteomics analyses, GO and KEGG analyses revealed that the most significantly perturbed pathways in both mutants were unrelated to flagellar function (Supplementary Fig. 6), indicating that TSSK6 and DUSP21 are involved in various cellular processes. Consistently, TSSK6 was reported to be required for Izumo relocalization and actin polymerization during the acrosome reaction, and for histone phosphorylation during spermiogenesis[27,31]. Given TSSK6's kinase activity and DUSP21's phosphatase function, we hypothesized that *Tssk6⁻/⁻* and *Dusp21⁻/y* spermatozoa would exhibit phosphorylation perturbation. We thus enriched the phosphopeptides using a phosphorylation enrichment column and cross-referenced the quantitative proteomics and phosphoproteomics data to exclude phosphorylation changes caused by variations in protein abundance. We found that the phosphorylation levels of 5870 phosphorylation sites across 2431 proteins in *Tssk6⁻/⁻* spermatozoa, and 1771 phosphorylation sites across 986 proteins in *Dusp21⁻/y* spermatozoa were altered, respectively (best localization probability ≥ 0.95, fold change ≥ 2, FDR < 0.05, Supplementary Fig. 6a-b). Interestingly, both knockouts exhibited increases and decreases in phosphorylation at different sites, indicating a global perturbation of the sperm phosphorylation network. These proteins are predominantly involved in cilia movement and organization, and are widely distributed among MIPs and external axonemal proteins (Fig. 6). In *Dusp21⁻/y* spermatozoa, an increased proportion of up-phosphorylated sites was observed, consistent with its function as a phosphatase. *Tssk6⁻/⁻* spermatozoa showed more extensive dysregulation of axonemal protein phosphorylation than *Dusp21⁻/y*, correlating with its more severe phenotype (Fig. 6g-h). Previous studies have shown that phosphorylation of flagellar axonemal proteins, such as the IDAf component IC138, can regulate ciliary motility[32–36]. Thus, TSSK6 may modulate sperm motility by controlling the phosphorylation states of structural flagellar proteins, although additional pathways, such as chromatin remodeling, may also contribute (Fig. 6l).

## Analyses of respiratory cilia and laterality

Next, we examined whether other motile cilia are impacted in KO mice. We found that all respiratory cilia were motile, but their beat frequency was reduced in *Tekt1⁻/⁻* mice (Supplementary Fig. 7a, Supplementary Video 2). The beat frequency remained normal in the other KO mice. This finding aligns with the observation that TEKT1 is present in both respiratory cilia and sperm flagella, whereas TEKT5, TSSK6, and DUSP21 are specifically expressed in sperm. None of the examined mice exhibits laterality defects, suggesting that these genes are either absent or play an indispensable role in nodal cilia (Supplementary Fig. 7b).

## Discussion

High-resolution cryo-EM structures of DMTs isolated from different cell types and species have revolutionized our understanding of ciliary proteins in axonemal assembly and their cell-specific diversity. However, the physiological roles of DMT-associated proteins remain poorly defined. To address this gap, we generated genetic KO mouse models targeting four sperm DMT-enriched proteins to evaluate the functional significance of DMT-associated tektins and enzymes. While *Tekt5⁻/⁻* and *Tssk6⁻/⁻* mice have been partially characterized[26,37], mouse models targeting *Tekt1* and *Dusp21* have not been reported before. To comprehensively elucidate the functional roles of these proteins, we employed a multi-scale investigative framework spanning from organismal to molecular resolution. Our results demonstrate how DMT-associated filamentous and enzymatic proteins govern sperm function through divergent mechanisms.

## Tektins exhibit diverse impacts on sperm function

TEKT1 and TEKT5 exhibit strikingly divergent phenotypic impacts on sperm function. *Tekt1⁻/⁻* mice are completely male infertile and display severely disturbed sperm motility. Respiratory cilia of *Tekt1⁻/⁻* mice also display reduced beat frequency, consistent with a published case in which a compound heterozygous variant of TEKT1 was associated with airway ciliary motility defects[23]. In addition, *Tekt1* knockdown in zebrafish resulted in laterality defects, while knockdown in flies caused decreased sperm counts and male sterility[23,38]. These results, together with our data in mouse sperm, suggest that TEKT1 is indispensable in diverse motile cilia. Cryo-EM showed that the assembly of tektin bundle is perturbed in *Tekt1⁻/⁻* spermatozoa despite the presence of DMTs, indicating that the tektin bundle is essential for flagellar motility. In contrast, *Tekt5⁻/⁻* mice display comparable fertility to WT, although spermatozoa exhibit mild phenotypes, including decreased progressive ability and increased bent flagella after capacitation. Our observations in *Tekt5⁻/⁻* mice are consistent with a recent study employing a different knockout strategy, which reported that *Tekt5⁻/⁻* mice remain fertile with slightly reduced sperm motility[37], although human *TKET5* variants have been associated with non-obstructive azoospermia[39] and siRNA-mediated knockdown of *Tekt5* in mouse testes was suggested to cause abnormal spermiogenesis[40]. Interestingly, ~8% *Tekt1⁻/⁻* spermatozoa exhibit coiled flagella and the percentage of coiled tails increases to ~12% after 90 min of in vitro capacitation, whereas the spermatozoa of WT and other KO mice almost do not show coiled flagella (Fig. 3c, Supplementary Fig. 4c). This implies that tektin filaments may provide essential elastic resilience to DMTs during vigorous flagellar bending, mirroring the mechanoprotective role of intermediate filament networks in other cellular contexts[41]. Direct evidence from our WB analysis demonstrated that TEKT1 and TEKT5 stabilize the DMTs since *Tekt1⁻/⁻* and *Tekt5⁻/⁻* flagellar axonemes are more readily disturbed under ultrasonic treatments. Taken together, these findings strongly support the indispensable role of tektin filaments in maintaining DMT integrity for motility, consistent with the evolutionarily conserved function of tektins as intermediate filament-like structural proteins that are resistant to harsh chemical treatments[24,42]. Furthermore, these results suggest that the progressive assembly of tektins – first TEKT1-4 and then TEKT5 – gradually enhances the integrity, physical properties, and motility of sperm flagella.

Notably, a previous cryo-electron tomography (cryo-ET) study reported flexible densities in the TEKT5 region of DMTs from *Tekt5⁻/⁻* spermatozoa and suggested potential compensation by other tektins[37]. In our study, high-resolution cryo-EM analyses showed that TEKT5 density was barely detected in *Tekt5⁻/⁻* DMT, and western blot

confirmed the loss of TEKT5 at the protein level. Differences between studies may reflect distinct experimental systems, resolutions, or analytical approaches, and further work will be required to reconcile these observations.

## DMT-associated enzymes regulate sperm function

Our results indicate that TSSK6 and DUSP21 function primarily as enzymatic regulators rather than just structural building blocks. Global proteomic profiling revealed no significant enrichment of dysregulated proteins in flagellar motility pathways. In contrast, phospho-proteomic analyses revealed significant phosphorylation changes in *Tssk6*[-/-] and *Dusp21*[-/y] spermatozoa, with altered phosphorylation of axonemal proteins broadly distributed in MIPs and other axonemal complexes.

Why do functional enzymes bind to the DMT? This could be a mechanism to enhance enzyme activity and specificity. By anchoring

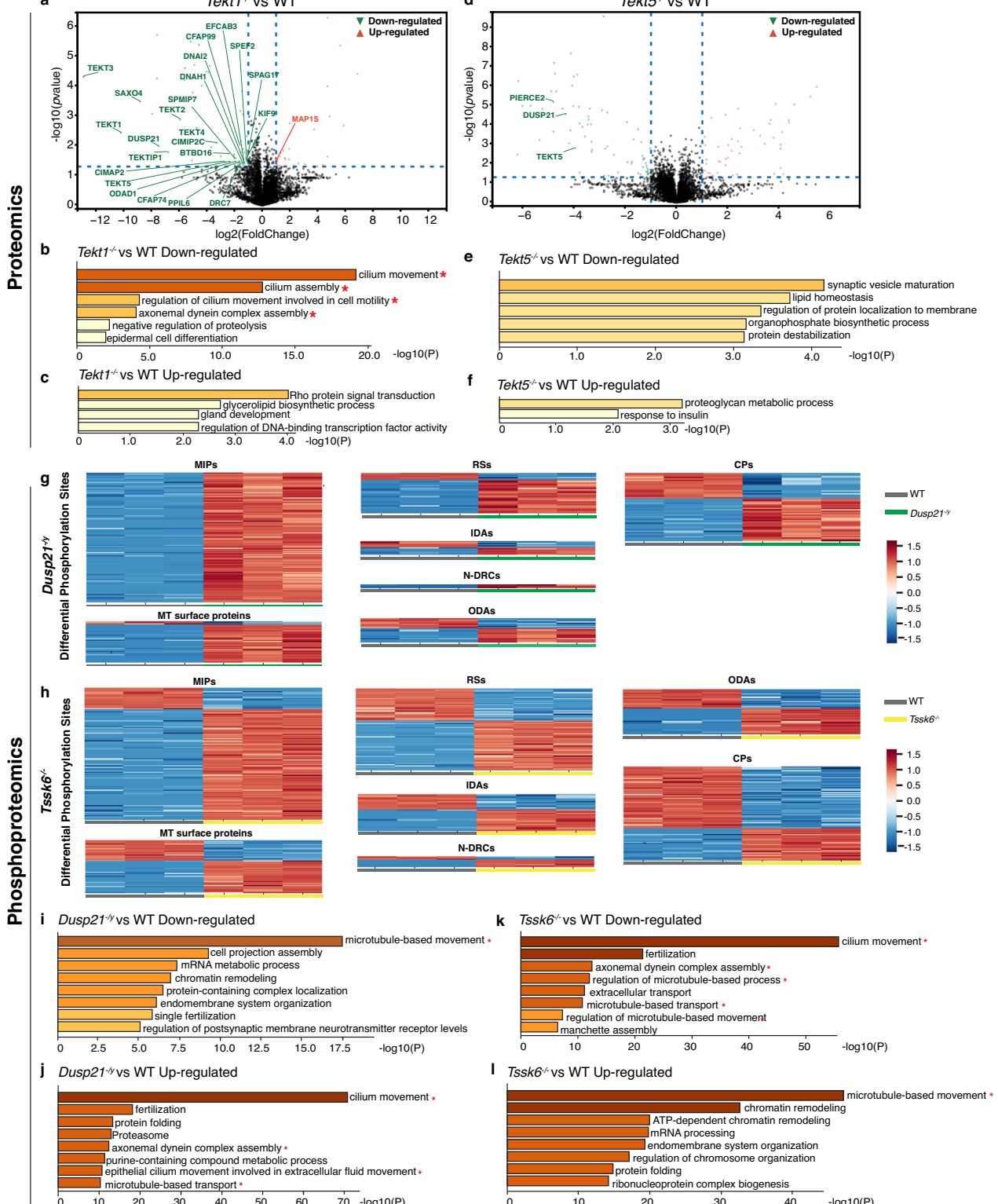

**Fig. 6 | Quantitative proteomics and phosphoproteomics between knockout and WT spermatozoa. a** Volcano plot shows the distribution of differentially expressed proteins (fold change ≥ 2, *p*-value < 0.05) in *Tekt1⁻/⁻* spermatozoa compared with WT. Red triangles represent up-regulated proteins, and green triangles represent down-regulated proteins. The differentially expressed axonemal proteins are marked. Statistical significance was assessed using an unpaired two-tailed *t*-test. **b**, **c** GO and KEGG analyses of down-regulated (**b**) and up-regulated (**c**) proteins in *Tekt1⁻/⁻* spermatozoa. GO and KEGG enrichments were obtained using the following parameters: min overlap = 3, *p*-value cutoff = 0.01, and min enrichment = 1.5. Statistical significance for term enrichment was assessed using a hypergeometric test, and *p*-values were adjusted for multiple testing using the Benjamini-Hochberg procedure to control the FDR. **d** Volcano plot shows the distribution of differentially expressed proteins in *Tekt5⁻/⁻* spermatozoa compared with WT. Statistical significance was assessed using an unpaired two-tailed *t*-test. **e**, **f** GO and KEGG analyses of down-regulated (**e**) and up-regulated (**f**) proteins in *Tekt5⁻/⁻* spermatozoa. The top five enriched terms were shown in (**e**). Statistical significance for term

enrichment was assessed using a hypergeometric test, and *p*-values were adjusted for multiple testing using the Benjamini-Hochberg procedure to control the FDR. **g**, **h** Heatmaps showing representative differentially phosphorylated sites (best localization probability ≥ 0.95, fold change ≥ 2, FDR < 0.05) in *Dusp21⁻/y* and *Tssk6⁻/⁻* spermatozoa. These phosphorylation sites are categorized based on known axonemal structural components, including MIPs, MT surface proteins, RSs, IDAs, N-DRCs, ODAs, and CPs. Statistical significance was assessed using an unpaired two-tailed *t*-test, and *p*-values were adjusted for multiple testing using the Benjamini-Hochberg procedure to control the FDR. **i**–**l** GO and KEGG analyses of down-regulated and up-regulated phosphorylated proteins in *Dusp21⁻/y* and *Tssk6⁻/⁻* spermatozoa. The top eight enriched GO terms were shown. Terms related to flagellar motility and assembly are marked with red asterisks. Statistical significance for term enrichment was assessed using a hypergeometric test, and *p*-values were adjusted for multiple testing using the Benjamini-Hochberg procedure to control the FDR. The source data underlying Fig. 6 are provided as a Source Data file.

to the DMT, TSSK6 and DUSP21 strategically position themselves near their axonemal substrates, which are distributed along the length of the flagellum. This localization restricts the accessibility of TSSK6 and DUSP21 to a more limited range of substrates, thus improving their enzymatic specificity. A comparable instance of subcellular localization influencing kinase specificity can be seen in the distinct locations of Cyclin B2 in the Golgi and Cyclin B1 in the cytoplasm and nucleus, which determine the mitotic events they regulate, independent of their inherent substrate specificity[43]. Furthermore, external axonemal complexes such as RS and CP also concentrate key enzymatic domains, including PP1c, PKA, ADK, and GUK[18,44]. These findings underscore the importance of the axoneme as a vital signaling hub for protein phosphorylation. Notably, DUSP21 stands out as the only mouse sperm MIP absent in human sperm DMT, whereas a homologous protein, DUSP3, is detected in the human sperm DMT by MS analysis[13], indicating a significant difference between species. Interestingly, the relatively mild phenotype observed in *Dusp21⁻/y* mutants suggests that other sperm phosphatases may compensate for the loss of DUSP21. Exploring how changes in phosphorylation states modulate flagellar motility would be an interesting topic in the future.

### Define TEKT1 as an MIVA gene and TSSK6 as a likely MMAF gene
Our previous study identified four male infertility patients harboring three different *TEKT1* variants[13]. Despite normal sperm morphology, these individuals exhibit asthenozoospermia, with progressive sperm motility falling below the WHO threshold of 32%[45]. Our *Tekt1⁻/⁻* mouse model mimics the human phenotype, as *Tekt1⁻/⁻* male mice exhibit infertility characterized by impaired sperm motility and altered waveform. Approximately 8% of *Tekt1⁻/⁻* sperm flagella are coiled, but no additional morphological defects are noted. Sperm count and length remain within normal ranges. Multiple Morphological Anomalies of the Sperm Flagella (MMAF) is a severe type of asthenozoospermia that exhibits absent, short, bent, coiled or irregular flagella readily identifiable under light microscopy[46]. These findings establish *TEKT1* as a pathogenic gene for non-MMAF asthenozoospermia, which was proposed as MIP-variant-associated asthenozoospermia (MIVA)[13].

In contrast, over 70% *Tssk6⁻/⁻* spermatozoa experience loss of flagella or exhibit bent tails, resembling the MMAF phenotype. Previously reported genes associated with MMAF are primarily axonemal genes involved in the assembly of ODA, IDA, N-DRC and RS structures[47]. Our findings indicate that enzymes such as TSSK6 may influence sperm morphology and motility, leading to male infertility. Notably, mutations of serine/threonine-protein kinase 33 (STK33) and adenylate kinase 7 (AK7) have been implicated in causing MMAF[48–50], indicating that defects in kinase-related phosphorylation may be a significant mechanism underlying MMAF. Further investigations are required to ascertain whether TSSK6 variants are present in patients with male

infertility. Additionally, flagellar regulators like kinases should be incorporated into the molecular diagnosis of male infertility.

### Diverse functions of axonemal proteins in different cells
Despite the evolutionarily conserved "9 + 2" microtubule architecture shared by motile cilia and sperm flagella, their molecular compositions exhibit marked divergence[12,13,16–18]. This partially explains why mutations of certain axonemal proteins cause tissue-specific phenotypes[47,51]. Consistently, testis-specific TSSK6 ablation in mice results in male infertility due to sperm motility defects, while respiratory and nodal cilia retain normal function, as demonstrated by the preserved coordinated beating of respiratory cilia and normal left-right body patterning. What's more, variants of commonly expressed genes in sperm flagella and motile cilia can also exhibit diverse disease outcomes. TEKT1 is found in both respiratory cilia and sperm flagella. Our investigation revealed that the depletion of TEKT1 moderately influences the motility of respiratory cilia, and the nodal cilia are likely to retain their function as evidenced by normal laterality (Supplementary Fig. 7). In comparison, this depletion severely disrupts the integrity and motility of sperm, implying that tektin bundles are more crucial for sperm function than for epithelial cells. This phenomenon has been observed in various axonemal proteins. For instance, mutations in CFAP61, an RS3 component, cause MMAF without manifesting any primary ciliary dyskinesia (PCD) symptoms[52,53], while mutations in ODAD1, an ODA-DC component, exhibit PCD with normal sperm function[54]. These varying effects of shared axonemal proteins in sperm flagella and motile cilia highlight the need for mechanistic studies to resolve how these proteins achieve functional specialization across cell types.

Our work highlights the power of cryo-EM and proteomics in providing an integrative framework for understanding axonemal protein function, as cryo-EM resolves structural perturbations at residue-level resolution, while proteomics captures global protein and phosphorylation dynamics. Future investigations incorporating these methodologies will be essential to comprehensively unravel the physiological functions of additional axonemal proteins. Together with clinical research, these efforts will help identify the causative genes and potential treatments for male infertility and ciliopathies.

## Methods
### Ethics statement
All animal studies and experiments were conducted in compliance with institutional guidelines and approved by the Institutional Animal Care and Use Committee (IACUC) at Westlake University (Hangzhou, China).

### Animals
All mice used in this study were on a C57BL/6 J background. *Tekt1⁻/⁻* mice (Strain NO. T030593), *Tekt5⁻/⁻* mice (Strain NO. T035937), *Dusp21⁻/y*

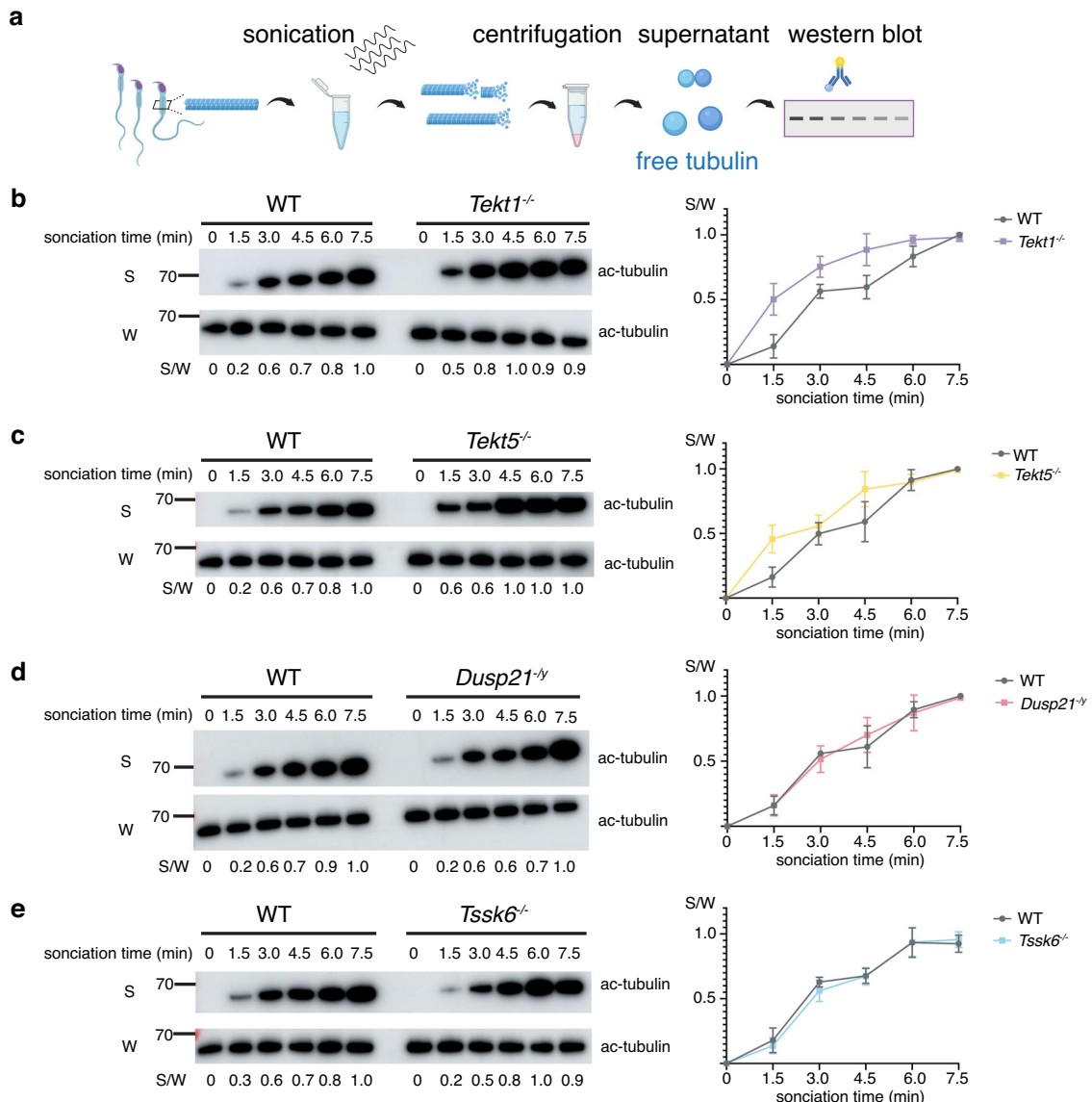

**Fig. 7 | Stability analyses of sperm DMTs from knockout mice. a** Cartoon diagram of the sperm stability assay. Spermatozoa were sonicated and centrifuged to separate supernatant fractions. Free acetylated tubulin levels in the supernatant were analyzed by western blot. Schematic diagram was was created with BioRender Gui, M. (2026) [https://BioRender.com/jhjkmh2]. **b–e** Western blots show the release of ac-tubulin following treatments with various sonication durations of sperm flagella from *Tekt1⁻/⁻* (**b**), *Tekt5⁻/⁻* (**c**), *Dusp21⁻/y* (**d**) and *Tssk6⁻/⁻* (**e**) mice, compared to WT. The supernatant-to-whole spermatozoa (S/W) ratios of ac-tubulin were recorded for each sonication duration (0, 1.5, 3, 4.5, 6, 7.5 min) below the respective western blot lanes. The S/W ratios were normalized by setting the largest number to 1 and the other numbers to their ratios relative to the largest. The line graph of the S/W ratios of ac-tubulin is displayed on the right. Three biological replicates were performed. The statistical data are presented as mean ± SD. The source data underlying Fig. 7 are provided as a Source Data file.

mice (Strain NO. T035080) and *Tssk6⁻/⁻* mice (Strain NO. T034984) were purchased from GemPharmatech (Nanjing, China). The mice were maintained in barrier facilities with a strictly controlled macroenvironment, including a temperature of 20-26 °C, humidity ranges of 40-70%, and a 12-hour light/12-hour dark cycle.

## Genotyping
Genotypes of each knockout mouse strain were confirmed by PCR with two pairs of primers. The design of primers was summarized in Supplementary Fig. 3a and sequences were provided in Supplementary Table 4.

## RNA extraction, cDNA synthesis, and RT-PCR
Total RNA was extracted from the testes of male mice using RNAsimple Total RNA Kit (TIANGEN). The first chain of cDNA was synthesized using HiScript III 1st Strand cDNA Synthesis Kit (+gDNA wiper) (Vazyme). The gene expression level was assessed via PCR using the Primer Pair (Beyotime) in corresponding knockout male mice with WT mice as a control and normalized with the expression of *Gapdh*.

## Isolation of mouse spermatozoa
Mouse spermatozoa were isolated from 2- to 4-month-old male mice. Mouse epididymides were dissected and immersed in pre-warmed Krebs-Ringer buffer (Solarbio). Spermatozoa were isolated via swimout or syringe-assisted epididymal flush strategies. For swim-out, the epididymides were cut into fragments and incubated at 37 °C for over 5 min to release spermatozoa. In the latter strategy, the epididymal lumen was flushed with 50 mL of buffer using a syringe prior to being cut into fragments to facilitate the collection of additional spermatozoa. Sperm suspensions were pooled and centrifuged at 1,500 × g for

10 min. The pellet was washed twice with Krebs-Ringer buffer to remove debris, resuspended in Krebs-Ringer buffer containing 30% glycerol, snap-frozen in liquid nitrogen, and stored at −80 °C.

## Western blotting

Epididymal spermatozoa from adult male mice were collected and resuspended in lysis buffer (8 M urea, 50 mM Tris-HCl, pH 8.2, 75 mM NaCl) containing 2 mM Phenylmethylsulphonyl fluoride (PMSF), followed by sonication for 30 s. After incubation on ice for 30 min, the supernatants were collected by centrifugation at 12,000 × g for 30 min at 4°C, mixed with loading buffer and then boiled at 95°C for 10 min. The proteins were separated by SDS-PAGE and transferred to PVDF membranes (Merck Millipore). The membranes were blocked with 5% skim milk in TBST (Tris-buffered saline with 0.1% Tween-20 detergent) at 4°C overnight. After being washed four times with TBST for 10 min, the membranes were incubated with the primary antibodies: rabbit anti-TEKT1 (1:1000, Bioworld), mouse anti-TEKT5 (1:2000)[16], rabbit anti-DUSP21 (1:1500, this study), rabbit anti-TSSK6 (1:1000, this study), mouse anti-β-actin (1:3000, Sigma) or mouse anti-acetylated tubulin (1:10000, Sigma) at 4°C overnight. The membranes were washed four times with TBST for 10 min and then incubated with anti-rabbit (Absin Bioscience) or anti-mouse (Abmart) HRP-conjugate secondary antibodies (1:5000) for 1 h at room temperature. The membranes were developed using a chemiluminescence kit (Beyotime) and the signal was captured by Amersham Imager 600 (GE Healthcare). The signal intensities were evaluated by Fiji software.

Rabbit polyclonal antibody generated against DUSP21 and Rabbit monoclonal antibody generated against TSSK6 were produced by HUABIO (Hangzhou, Zhejiang, China). The synthesized peptide of DUSP21 (CIFPSQATQQDNIY) conjugated to KLH and the purified full-length protein of TSSK6 were used to immunize rabbits.

## Analyses of testis and epididymis histology and sperm flagellar length

Testes and epididymides were fixed in Bouin's solution (Sigma) and embedded in paraffin wax. The paraffin sections were deparaffinized, rehydrated and stained with hematoxylin and eosin (H&E, HaoKe Biotechnology). To measure the sperm flagellar length, the cauda epididymal spermatozoa were dispersed in Human Tubal Fluid (HTF) medium (FUJIFILM Irvine Scientific) on the slides and images were collected on an Olympus IX-83 microscope equipped with a DP80 camera. Flagellar lengths of over 100 spermatozoa each mouse were measured using Fiji software.

## Mating test

Each *Tekt5*[-/-] or *Dusp21*[-/y] adult male mouse was caged with one female mouse, while each WT control was caged with one female mouse. Each *Tekt1*[-/-] or *Tssk6*[-/-] adult male mouse was caged with two female mice, whereas each WT control was caged with two female mice simultaneously. All groups were housed for over three months, and the average litter size was calculated at the end of the experiments.

## In vitro fertilization

3-4 weeks female mice were superovulated by injecting 10 U of Serum Gonadotropin for Injection (Sansheng Biological Technology) and after 46-48 hours injecting 10 U of Chorionic Gonadotrophin for Injection (Sansheng Biological Technology). After 16-18 hours, cumulus-oocyte complexes (COCs) were collected. Epididymal sperm from *Tekt1*[-/-] or *Tssk6*[-/-] male mice, with WT male mice as a control, were capacitated in TYH (Toyoda, Yokoyama, Hoshi) medium at 37°C under 5% CO₂ for about 1 hour and then inseminated to the COCs. After 6-hour incubation, the oocytes were washed twice in HTF medium and cultured in fresh HTF overnight at 37°C under 5% CO₂. 2-cell embryos were counted 22-24 h after insemination and 2-cell% was calculated.

## Sperm morphology analysis

For sperm morphological analysis, epididymal spermatozoa from one cauda epididymis were collected in Krebs-Ringer buffer within 5 min, termed 0-min no-capacitated spermatozoa. Epididymal spermatozoa from the other cauda epididymis from the same mouse were incubated in HTF medium (FUJIFILM Irvine Scientific) containing 10% Fetal Bovine Serum (FBS) at 37°C under 5% CO₂ for 90 min, termed 90-min capacitated spermatozoa. Movies were recorded for 1 second with 200 frames per second (fps) at 37°C using a Nikon Spinning Disk CSU-W1 SoRa microscope equipped with camera ORCA-Fusion BT (HAMAMATSU). Spermatozoa were classified per 6[th] WHO laboratory manual for the examination and processing of human semen.

## Sperm motility and waveform analysis

Fresh epididymal spermatozoa were incubated in HTF medium containing 10% FBS at 37°C under 5% CO₂. The sperm motility was assessed with Hamilton Thorne's Ceros II system (Hamilton-Thorne Research) by acquiring 45 frames at a frame rate of 60 Hz. Over 300 spermatozoa were analyzed at 5 min or 90 min each time.

For flagella waveform analysis, epididymal spermatozoa were collected in Krebs-Ringer buffer within 5 min or in EmbryoMax® Human Tubal Fluid (HTF) (Sigma) at 37°C under 5% CO₂ for 90 min, respectively. Confocal Dish (Coverglass Bottom Dish, Solarbio) was coated with fibronectin (Corning) to tether the sperm head for a planar beating. Within 10 min after transferring sperm to the coated dish, the sperm flagellar movements were recorded for 1 second with 200 frames per second (fps) at 37°C using a Nikon Spinning Disk CSU-W1 SoRa microscope equipped with camera ORCA-Fusion BT (HAMAMATSU). A complete beating cycle was chosen to generate overlaid images of flagellar waveform traces with time-coded in color using FIJI software. The beating frequency (the numbers of beating cycles per second) was measured.

## Cross-sectional TEM

Spermatozoa were fixed overnight at 4°C using 2.5% glutaraldehyde (Solarbio). Samples were rinsed twice with 0.1 M PBS for 10 min each, then fixed with osmium tetroxide for 1 hour. After three rinses with distilled water for 10 min each, samples were stained with 2% uranyl acetate for 30 min. Dehydration was achieved through gradient ethanol washes: 50%, 70%, 90%, and 100%, followed by two final rinses with 100% acetone for 15 min each. Samples were embedded in epoxy resin and polymerized at 60°C overnight. Ultrathin sections were prepared using a Leica UC7 ultramicrotome, collected on copper grids, and imaged with FEI Tecnai G2 120 kV transmission electron microscopy.

## Immunofluorescence staining

Spermatozoa were smeared onto poly-lysin coated slides (Sigma), air-dried, and fixed with 4% paraformaldehyde for 30 min at room temperature. Following three 5-min washes with PBST (phosphate buffered saline containing 0.1% Tween-20), samples were permeabilized in PBS buffer with 0.5% Triton X-100 at 37°C for 10 min, and washed three times with PBST for 5 min each. Slides were blocked with 5% bovine serum albumin (BSA) in PBST for 1 hour and then incubated overnight at 4°C with the following primary antibodies diluted in PBST containing 1% BSA and 0.3% Triton X-100. Primary antibodies included: mouse anti-acetylated tubulin (1:200, Sigma), rabbit anti-RSPH9 (1:100, Invitrogen), rabbit anti-DNAH1 (1:200, Invitrogen), rabbit anti-DNAH17 (1:200, Proteintech), rabbit anti-CFAP21 (1:200, Invitrogen), rabbit anti-SPAG6 (1:100, Sigma). After being washed with PBST three times, the slides were incubated for 1 hour with goat anti-rabbit or anti-mouse secondary antibodies conjugated to Alexa Fluor 488 or Alexa Fluor 647 (1:200, Abcam) and nucleus were counterstained for 10 min with DAPI (1:200, Beyotime) in the dark. The slides were washed with PBST and the samples were preserved in mounting media (anti-fade mounting media, Beyotime) to prevent photobleaching. The slides were sealed

with fingernail polish. Imaging was performed using confocal laser scanning microscopy (Olympus FV3000) equipped with the FV31S-SW software platform.

## Sperm flagella stability

Frozen spermatozoa were thawed in a 37 °C water bath and centrifuged at 1500 × g for 10 min. To remove membranes, spermatozoa were resuspended with Krebs-Ringer buffer supplemented with 0.3% Triton X-100 at room temperature for 30 min, followed by centrifugation at 2500 × g for 8 min to collect axonemes. Axonemal fractions were subjected to non-contact ultrasonication (XIAO-MEICHAOSHENG, XM-26A) for 1.5, 3.0, 4.5, 6.0, and 7.5 min at 50% amplitude, then centrifuged at 20,000 × g for 10 min. Supernatants and whole spermatozoa were analyzed by western blot using an anti-acetylated α-tubulin antibody (1:10000, Sigma) to quantify free acetylated tubulin. Images were processed by ImageJ to quantify the intensity of bands. When calculating the ratio of tubulin in the supernatant to that in whole cells, the whole-cell tubulin value was determined as the average signal across different sonication times within the same genotype. The S/W ratios were normalized by setting the largest number as one and the other numbers as the ratio compared with the largest number. The experiment was repeated three times independently.

## Laterality and tracheal ciliary beat frequency (CBF) analyses

2- to 4-month-old male mice were euthanized by cervical dislocation and secured in a supine position on a dissection tray. A midline thoracic incision was made from the xiphoid to the mandible in an inverted "T"-shaped pattern to fully expose the thoracic cavity. To assess organ laterality, a midline incision was made along the ventral abdominal surface to fully expose the thoracic and abdominal cavities. The organ positions were documented to assess situs anomalies, including abdominal organs (heart and lungs) and abdominal organs (liver, stomach, spleen, and intestinal rotation patterns).

For tracheal CBF analysis, the "lungs + bronchi + trachea (up to the epiglottis)" were excised and placed in pre-warmed PBS. Connective tissues, esophagus, and blood vessels surrounding the trachea and lungs were carefully removed using ophthalmic forceps. The trachea was separated from the left and right lungs by dissecting along the left and right main bronchi. The trachea was rinsed thoroughly in fresh PBS to remove blood and debris and was transferred to pre-warmed M199 medium (Gibco). Using ophthalmic scissors, the trachea was opened longitudinally (perpendicular to the cartilage rings) from the epiglottis, bisected into strips, and the middle-to-lower tracheal segments (rich in motile cilia) were trimmed into 2 × 2 mm pieces. Tracheal fragments were placed lumen-side down in a glass-bottom dish (Biosharp). A drop of M199 medium was added, and the sample was covered with a coverslip. The dish was positioned on the stage of a live-cell spinning disk confocal microscope (Olympus IX83 equipped with a high-speed camera). Samples were maintained at 37 °C with 5% $CO_2$. The ciliary beating was recorded for 9 s at 200 fps. Videos were processed using Fiji software to quantify ciliary beat frequency. We set multiple lines across a beating cilium in the video image and measured the wave of the light intensity change. The distances from the top of the peaks to the bottom were measured over five waves. The number of peaks and the averaged distance were used to determine the CBFs.

## Preparation of mouse DMTs

Sample preparation and data collection of DMTs from WT mouse were described previously[13]. Epididymal spermatozoa of KO mice were resuspended to ~1 × 10^7 cells/mL in 1.5 mL HMEKDP buffer (30 mM HEPES, pH 7.5, 5 mM MgSO_4, 0.5 mM EDTA, 50 mM KCl, 1 mM DTT,

1 mM PMSF) and then supplemented with 0.2% Triton X-100 detergent (Sigma) for 15 to 30 min at room temperature. After centrifugation at 1500 × g for 5 min, the pellet was washed twice with HEDMPK buffer. The axoneme was resuspended in HEMK buffer (10 mM HEPES, pH 7.2, 0.5 mM EDTA, 5 mM MgSO_4, 50 mM KCl) supplemented with 10 mM ATP, 10 mM MgCl_2 and 750 μM CaCl_2 for 15 min at room temperature to splay into DMTs. The DMTs were collected by centrifugation at 6000 × g for 5 min and resuspended in HMEK buffer for cryo-EM grid preparation.

## Cryo-EM sample preparation and data acquisition

A 3.5 μL aliquot of the mouse sperm DMT sample was loaded onto glow-discharged holey carbon grids (Quantifoil Au R2/1, 200 mesh or 300 mesh, or Quantifoil Au R1.2/1.3, 300 mesh). The grids were blotted for 4 s and then plunged frozen in liquid ethane cooled by liquid nitrogen using Vitrobot Mark IV (Thermo Fisher Scientific) at 100% humidity and 8 °C.

Cryo-EM data of *Tekt5*[-/-] DMT were collected in the cryo-EM facility of Westlake University at a nominal magnification of 81,000× on a Titan Krios (Thermo Fisher Scientific) operating at 300 kV equipped with a K3 Summit detector and GIF Quantum energy filter (slit width 20 eV) in super-resolution mode. Movie stacks were automatically acquired using EPU (Thermo Fisher Scientific). The defocus range was set from −1.5 to −2.6 μm. Each movie stack, consisting of 32 frames, was exposed for 2.56 s with a total dose of ~50 e⁻/Å². The super-resolution movies were downscaled by a factor of 2 for subsequent data processing, yielding a pixel size of 1.087 Å.

Cryo-EM data of *Tekt1*[-/-], *Dusp21*[-/y] and *Tssk6*[-/-] DMT were collected in the cryo-EM facility of Liangzhu Laboratory. Movie stacks were collected with EPU on a Titan Krios G4 (Thermo Fisher Scientific) operating at 300 kV equipped with a Falcon 4i detector and Selectris X energy filter (slit width 20 eV) with a pixel size of 0.93 Å. The defocus range was set from −1.5 ~ −2.5 μm and the total dose per movie stack was ~50 e⁻/Å².

## Image processing

Cryo-EM data of WT and KO DMTs were processed using a similar strategy in RELION[55] and were summarized in Supplementary Figs. 1 and 5, respectively. Movie stacks were motion corrected by RELION's implementation of motion correction and CTF parameters were calculated by CTFFIND4.1[56]. After pre-processing, particle picking was performed using multiple approaches, including filament tracer in cryoSPARC[57], crYOLO[58] and manual picking. The selected DMTs were segmented into boxes spaced 8 nm apart, referred to as 8-nm particles, and all particles were extracted and subjected to an initial 8-nm 3D refinement with a density map of mouse sperm DMT (EMD-35823) as the initial model. The following 16-nm and 48-nm 3D classifications were performed with cylinder masks on CFAP52 (16-nm repeat) and NME7 (48-nm repeat), respectively. The 48-nm particles were subjected to Bayesian polishing before being used for 3D refinement to generate a 48-nm map. For WT maps, local masked refinements were performed to further improve map quality.

## Model building

Proteins are summarized in Supplementary Fig. 2. Model building was performed in Coot[59] with torsion, planar peptide, trans peptide, and Ramachandran restraints applied. To model the MIPs in mouse sperm DMT, the structures of homologous proteins in bovine sperm DMT (PDB ID: 8OTZ) were fitted into cryo-EM density using UCSF ChimeraX[60], and the models were then substituted by corresponding mouse homology models generated in SWISS-MODEL[61] or AlphaFold2[62] and refined against their corresponding cryo-EM density. The model was adjusted to fit the density. The atomic models of mouse and human sperm DMTs were refined using Phenix.real_space_refine with secondary structure, Ramachandran and rotamer restraints

applied[63]. Model refinement statistics were calculated in Molprobity integrated into Phenix. Models and density maps were visualized in UCSF ChimeraX.

## Quantitative proteomics sample preparation

The quantitative proteomics analyses were conducted in three independent experimental groups, each comprising WT controls and corresponding knockout samples. Group 1 included 9 samples from 3 biological replicates of *Dusp21^(-/y)^*, 3 biological replicates of *Tssk6^(-/-)^*, and 3 biological replicates of WT. Group 2 consisted of 6 samples from 3 biological replicates of *Tekt1^(-/-)^*, and 3 biological replicates of WT. Group 3 consisted of 6 samples from 3 biological replicates of *Tekt5^(-/-)^*, and 3 biological replicates of WT. Each replicate was obtained from an individual mouse. No technical replicates were performed for LC-MS/MS runs.

Epididymal spermatozoa from *Tekt1^(-/-)^*, *Tekt5^(-/-)^*, *Dusp21^(-/y)^*, *Tssk6^(-/-)^* and WT adult male mice were collected and lysed with RIPA (Beyotime) supplemented with protease inhibitor (Roche) and sonication (Bioruptor Pico, Diagenode). The supernatant was collected after centrifugation at $16,000 \times g$ for 10 min at 4°C and the concentration of the extracted proteins was determined using a bicinchoninic acid (BCA) assay (Thermo Scientific). A quantity of 20 μg of total protein was reduced with 10 mM dithiothreitol (DTT) at 55 °C for 45 min, followed by alkylation with 50 mM iodoacetamide at room temperature in the dark for 30 min. Purification was performed employing SP3 technology as described by Hughes et al. [64]. The purified proteins were digested with trypsin (Promega) using a 1:100 enzyme-to-protein ratio (w/w) at 37 °C overnight. The resulting peptides were then lyophilized in preparation for LC-MS/MS analysis.

## Quantitative proteomics LC-MS/MS data acquisition

For LC-MS/MS analysis, the resulting peptides were separated using a Thermo Vanquish Neo integrated nano-HPLC system with a flow rate of 300 nL/min in a 125-min gradient elution. The system was directly interfaced with the Thermo Exploris 480 mass spectrometer. The analytical column was a homemade fused silica capillary column (75 μm inner diameter, 150 mm length; Upchurch) packed with C-18 resin (300 A, 3 μm; Varian). Mobile phase A consisted of 0.1% FA in water, and mobile phase B consisted of 80% ACN and 0.1% FA. The mass spectrometer was operated in the DDA mode using the Xcalibur 4.1 software. Each cycle consisted of a single full-scan mass spectrum in the Orbitrap (400-1800 m/z, 60,000 resolution), followed by 20 data-dependent MS/MS scans (300-1500 m/z, 15,000 resolution) at 30% normalized collision energy. The automatic gain control (AGC) target was set at 5e4, and the maximum injection time was 50 milliseconds. Each mass spectrum was analyzed using the Thermo Xcalibur Qual Browser.

Label-free quantification and protein identification were conducted using Proteome Discoverer 2.5 for database searching. Searches were performed against the Mus musculus proteome database downloaded from UniProtKB (UP000000589). The Sequest search parameters included a 10-ppm precursor mass tolerance, 0.02 Da fragment ion tolerance, and up to 2 internal cleavage sites, a minimum of 4 peptide segments, and a minimum of 1 unique peptide. Fixed modifications included cysteine alkylation, and the methionine oxidation was a variable modification. Peptides were filtered with a 1% false discovery rate (FDR).

## Phosphoproteomics sample preparation

The phosphoproteomics analyses included 9 samples from 3 biological replicates of *Dusp21^(-/y)^*, 3 biological replicates of *Tssk6^(-/-)^*, and 3 biological replicates of WT, with each replicate derived from an individual mouse. No technical replicates were performed for LC-MS/MS runs.

Proteins were extracted using an 8 M Urea buffer, and the concentration of the extracted proteins was determined using a BCA assay (Thermo Scientific). A quantity of 300 μg of total protein was reduced with 5 mM dithiothreitol at 30 °C for 60 minutes, followed by alkylation with 20 mM iodoacetamide at room temperature in the dark for 30 minutes. Ensure that the urea concentration was diluted to below 1.5 M, and that the proteins were digested with trypsin and lysC using a 1:100 enzyme-to-protein ratio (w/w) at 30 °C overnight. After enzymatic hydrolysis, the peptide segments were desalted with the Desalting Tip (Shanghai Omicsolution, OSFP0200-W). The purified peptides were subjected to phosphorylation enrichment with the EnrichmentTip (Shanghai Omicsolution, OSFP0005). The resulting peptides were lyophilized in preparation for LC-MS/MS analysis.

## Phosphoproteomics LC-MS/MS data acquisition

For LC-MS/MS, lyophilized peptides were resuspended in 20 μL of 0.1% formic acid and 3 μL aliquots were injected using the nanoElute UHPLC System (Bruker) onto a 25 cm×75 μm ID, 1.6 μm C18 column (Aurora series, IonOpticks). Peptides were eluted with a gradient of water/0.1% formic acid (A) and acetonitrile/0.1% formic acid (B) over 60 minutes at a flow rate of 300 nL/min, with the linear gradients starting from 2% B and increasing to 22% in 45 min, followed by an increase to 35% B in 50 min, 80% B in 55 min, 80% B in 55–60 min. Eluted peptides were analyzed with a TIMS quadrupole time-of-flight timsTOF Pro 2 instrument (Bruker Daltonics) using a CaptiveSpray nano-electrospray source. The dda-PASEF mode was employed with a m/z range of 300 to 1500 and a 1/K0 range of 0.75-1.3, with a Ramp Time of 166 ms.

Label-free quantification was conducted using MSFragger via FragPipe v22.0, MSFragger 4.1, and Philosopher 5.1.1. The LFQ-MBR workflow was employed with modifications appropriate to the experiment[65,66]. Searches were performed against the Mus musculus proteome database downloaded from UniProtKB (UP000000589). Search parameters matched those of the protein identification step, with IonQuant settings including a mass tolerance of 20 ppm, a retention time tolerance of 3 minutes, and an ion mobility tolerance of 0.05. Normalization was enabled, and a minimum isotope count was set to 2. Fixed modifications included cysteine alkylation, and the methionine oxidation, STY phosphorylation was variable modification. Peptides were filtered at 1% FDR. 42,199 peptides and 26,611 phosphorylation sites in 4,915 proteins were detected in *Dusp21^(-/y)^* spermatozoa, while 47,891 peptides and 33,407 phosphorylation sites in 5960 proteins were detected in *Tssk6^(-/-)^* spermatozoa.

## Quantitative proteomics and phosphoproteomics analyses

Mass spectrometry intensity values were preprocessed by replacing zero/missing values with a constant that is close to the minimum intensity value to avoid undefined logarithmic transformations. All values were log2-transformed to stabilize variance and normalize distributions. For each protein, the fold change (FC) between the KO and WT groups was calculated as the difference in mean log2 intensities from three biological replicates. Statistical significance was assessed using a two-tailed, two-sample Student's t-test assuming equal variances. Proteins with $|\log2(FC)| \geq \log2(2)$ and a $p$-value < 0.05 were defined as differentially expressed proteins.

Phosphorylation site intensities were processed similarly to the proteomics workflow, including zero-value imputation, log2 transformation, and statistical analysis (best localization probability ≥ 0.95, fold change ≥ 2, FDR < 0.05). To distinguish phosphorylation changes driven by protein abundance variations, significantly altered phosphosites were cross-referenced with the quantitative proteomics dataset. Phosphosites whose changes correlated with corresponding protein expression shifts were excluded, ensuring that only phosphorylation-specific regulatory events were retained. When the phosphorylation levels of multiple peptides in one protein were altered, the largest value was used for analysis.

The GO and KEGG enrichment analyses of differentially expressed proteins were performed using an online server (https://metascape.org). Pathway and Process Enrichment analysis was conducted with the

following parameters: minimum overlap 3, *p*-value cutoff 0.01, minimum enrichment 1.5. Enriched terms were derived from the GO biological processes and KEGG pathway database.

Differentially regulated phosphorylation sites of axonemal proteins were used to generate phosphorylation sites heatmaps. The intensity values of phosphorylated peptides were first log2-transformed and then standardized by Z-score normalization across samples. Heatmaps were generated using the seaborn package in Python, with a blue-white-red color scale representing relative abundance.

## Statistics and reproducibility
Statistical analyses were performed using two-tailed unpaired Student's t-test or one-way analysis of variance (ANOVA) in Prism. For phosphoproteomics analyses, *p*-values were adjusted for multiple testing using the Benjamini-Hochberg procedure to control the false-discovery rate (FDR). For GO/KEGG analyses, statistical significance for term enrichment was assessed using a hypergeometric test, and *p*-values were adjusted for multiple testing using the Benjamini-Hochberg procedure to control the FDR.

## Reporting summary
Further information on research design is available in the Nature Portfolio Reporting Summary linked to this article.

## Data availability
Cryo-EM maps have been deposited in the Electron Microscopy Data Bank with the accession numbers EMD-64679 for WT mouse sperm DMT, EMD-64623 for *Tekt1* KO mouse sperm DMT, EMD-64624 for *Tekt5* KO mouse sperm DMT, EMD-64625 for *Tssk6* KO mouse sperm DMT, EMD-64626 for *Dusp21* KO mouse sperm DMT. The corresponding refined atomic model has been deposited in the Protein Data Bank under accession number 9V10 for WT mouse sperm DMT. Previously published structures used in this study are available in the Protein Data Bank under the following accession codes 8IYJ (mouse sperm DMT), 8OTZ (bovine sperm DMT), 7RRO (bovine trachea DMT), 9CPB (bovine oviduct DMT), 8J07 (human trachea DMT), 8SNB (sea urchin sperm DMT). Previously published cryo-EM maps used in this study are available in the Electron Microscopy Data Bank with the accession numbers EMD-35810 (human sperm DMT) and EMD-45785 (human oviduct DMT). Mass spectrometry data have been deposited in ProteomeXchange with identifier PXD-073076 [https://proteomecentral.proteomexchange.org/cgi/GetDataset?ID=PXD073076]. All data supporting the results of this study are provided in the article, supplementary information, and Source Data file. Source Data are provided within the Source Data File. Source data are provided with this paper.

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

## Acknowledgements

We thank Dr. Alan Brown, Dr. Hannah Mitchison, and Dr. Radhika Subramanian for their critical suggestions. We thank Yi Zhu for the valuable suggestions on MS data analyses. We thank the Cryo-EM Facility of Liangzhu Laboratory and the Cryo-EM Facility and High-Performance Computing Center of Westlake University for the support on data collection and computation; Xiaoxia Wan in the center of Cryo-EM at Zhejiang University for her technical assistance on TEM; the Laboratory Animal Resources Center (LARC) of Westlake University for mouse maintenance and Jinwu Dai in assistance with the IVF experiments; the Microscopy Core Facility of Westlake University for technical assistance; the Mass Spectrometry & Metabolomics Core Facility of Westlake University for MS analysis; and the Core Facility of Liangzhu Laboratory for technical assistance. This work was supported by the National Key Research and Development Program of China (2025YFA1309900), the National Natural Science Foundation of China (NSFC) (32471245), the Zhejiang Provincial Natural Science Foundation of China (LZ24C050001 and LR26C050002), the Zhejiang Provincial Leading Innovation and Entrepreneurship Team Introduction and Cultivation Program (2024R01024), the Excellent Youth Science Fund of NSFC (Overseas) to M.G.; the National Natural Science Foundation of China (32271261), the Special Fund of the State Key Laboratory of Gene Expression (SKLGE-ZX-2025006), the Zhejiang Key Laboratories Project (2024E10052), Zhejiang Provincial Natural Science Foundation of China (LR22C050003 and LDG25C050002), Westlake University, and Institutional Startup Grant from the Westlake Education Foundation to J.W.

## Author contributions

M.G. and J.W. conceived the project. L.Z., X.L. generated KO mice, performed phenotype and sperm motility analyses, prepared cryo-EM samples and collected WT and *Tekt5* KO cryo-EM data. Q.L. collected *Tekt1*, *Tssk6* and *Dusp21* KO cryo-EM data and generated density maps, performed TEM, IF, MS, sperm stability, tracheal cilia and laterality analyses. M.G. and Q.L. built the atomic model. P.C., B.L., S.Y. contributed to mouse sperm isolation and particle picking. P.C. contributed to the stability experiment. B.L. conducted TEM. Y.L., H.Z., and S.X. contributed to the tracheal cilia and laterality experiments. J.H. and S.F. contributed to the collection and analysis of proteomic data. M.G., J.W., Q.L. and L.Z. designed the experiments and wrote the manuscript with input from all co-authors.

## Competing interests

The authors declare no competing interests.
