## [Transparent Peer Review file · Nature Communications]

Doublet Microtubule-Associated Tektins and Enzymes Differentially Regulate Sperm Flagellar Integrity and Motility

Corresponding Author: Professor Miao Gui

Version 0:

Reviewer comments:

Reviewer #1

(Remarks to the Author)

The authors have addressed all questions and suggestions raised in the previous round of review. The revised manuscript improved in clarity and completeness. The supplementation, adjustments to relevant content, and clarification of issues are appropriate.

Reviewer #2

(Remarks to the Author)

The authors did not address the previous comments. About single-particle EM data, if only a small portion (<10%) of the data led to the structural reconstruction, only those small portions of particles are certain to have a phenotype but simply stating knocking out xxx genes leads to certain phenotypes is scientifically misleading. Stating limitations and hypotheses of how previous papers should not be the focus of the current paper: first of all, the authors could be wrong; second, it does not address the limitation of the current paper.

Also the supplementary figure on microtubule stability is inconsistent for the control across panels and if the controls are swapped then most conclusions would not stand. However, it seems that within each experiment, the variation is incredibly small.

Reviewer #3

(Remarks to the Author)

In the revised manuscript, the authors have addressed all the points raised by the original Reviewer #1. It is now clear which parts of the study is novel compared with the previous related studies. This reviewer recommends the manuscript for publication in Nature Communications.

Reviewer #4

(Remarks to the Author)

REVIEWERS COMMENTS

Reviewer #1 (Remarks to the Author):

The authors have addressed all questions and suggestions raised in the previous round of review. The revised manuscript improved in clarity and completeness. The supplementation, adjustments to relevant content, and clarification of issues are appropriate.

We thank the Reviewer for noting that the revised manuscript has addressed all concerns.

Reviewer #2 (Remarks to the Author):

The authors did not address the previous comments. About single-particle EM data, if only a small portion (<10%) of the data led to the structural reconstruction, only those small portions of particles are certain to have a phenotype but simply stating knocking out xxx genes leads to certain phenotypes is scientifically misleading. Stating limitations and hypotheses of how previous papers should not be the focus of the current paper: first of all, the authors could be wrong; second, it does not address the limitation of the current paper.

We apologize for the lack of clarity. Although microtubule inner proteins in sperm DMTs exhibit a 48-nm periodicity, extracting particles at 48-nm spacing is impractical because it requires an excessively large alignment search range and leads to inefficient processing. Therefore, consistent with common practice in the field, we extracted DMT particles at 8-nm spacing and used 3D classification to recover the 16-nm and 48-nm repeats. In theory, 3D classification targeting the 16-nm feature yields two equivalent particle classes related by an 8-nm shift, of which one is selected (**Rebuttal Figure 1**). Subsequent classification targeting the 48-nm feature yields three equivalent classes related by 16-nm shifts, resulting in a theoretical particle retention rate of ~17% assuming no particle rejection. In practice, low-quality particles are removed from both WT and KO datasets, yielding observed retention rates of 5–9%. Importantly, the retention rates of the KO datasets are comparable to that of the WT dataset (7%), indicating that the reduced particle number reflects the expected outcome of the classification strategy rather than a KO-specific bias. Thus, a substantial fraction of the retained KO particles genuinely represent the observed phenotypes.

Recognizing the limitation that only a subset of particles contributed to the final 3D reconstructions, we have toned down the structural claims. In particular, definitive statements such as “proteins are absent” were revised to more conservative descriptions. Example clarifications include: “densities of TEKT5 and

its interacting partners – DUSP21, the C-terminus of TEX37, the N-terminal tail of TEKT3, and fragments of FAM166A – were barely observed", " Notably, proteins down-regulated in *Tekt1*^{-/-} spermatozoa, including TEKT1-5, TEKTIP1, DUSP21, SAXO4, and CIMIP2C (FAM166C), were also perturbed in the cryo-EM structure", " In *Dusp21*^{-/-} spermatozoa, cryo-EM analysis revealed largely diminished density of DUSP21 and its interacting partner". Furthermore, we have acknowledged the limitations of this approach and made it clear in the Results section that "Notably, near-intact external axonemal structures do not necessarily mean that all structures remain unchanged, as our cryo-EM image processing excluded some poor-quality particles."

Rebuttal Figure 1. (a) A theoretical schematic for recovering 48-nm particles with full particle retention. (b) A practical strategy for recovering 48-nm particles of *Tekt5*^{-/-} sperm DMTs. Classes with dashed boxes contain equivalent particles with 8-nm or 16-nm shifts. (c) Percentage of particles retained after 3D classification compared with the theoretical value.

Regarding the discussion of previous studies, we appreciate the reviewer's concern. Our intention in discussing points of consistency and inconsistency between our *Tekt5*^{-/-} mouse and previous studies was

not to speculate on potential limitations of prior work, but rather to provide context for interpreting our findings. We have revised the text to ensure that this discussion is descriptive rather than evaluative and that it does not detract from the focus of the current study. The revised discussion is provided below:

"Notably, a previous cryo-electron tomography (cryo-ET) study reported flexible densities in the TEKT5 region of DMTs from *Tekt5*^{-/-} spermatozoa and suggested potential compensation by other tektins. In our study, high-resolution cryo-EM analyses showed that TEKT5 density was barely detected in *Tekt5*^{-/-} DMT, and western blot confirmed the loss of TEKT5 at the protein level. Differences between studies may reflect distinct experimental systems, resolutions, or analytical approaches, and further work will be required to reconcile these observations."

Also the supplementary figure on microtubule stability is inconsistent for the control across panels and if the controls are swapped then most conclusions would not stand. However, it seems that within each experiment, the variation is incredibly small.

To calculate the ratio of tubulin in the supernatant relative to whole cells (S/W), the whole-cell tubulin value was defined as the average signal obtained across different sonication times within the same genotype. The S/W ratios were then normalized by setting the largest value to 1 and scaling all other values relative to it. This approach compares relative changes between experimental and control groups; therefore, multiplying all control blot intensities by a constant factor would not alter the conclusions. To verify this, we regrouped all 12 WT control blots into a single reference line. The resulting analysis is consistent with the line graphs shown in **Fig. 7b–e (Rebuttal Figure 2)**.

Rebuttal Figure 2. The line graphs of the S/W ratios of ac-tubulin in which all 12 WB blots of the WT control are grouped as a single line.

Reviewer #3 (Remarks to the Author):

In the revised manuscript, the authors have addressed all the points raised by the original Reviewer #1. It is now clear which parts of the study is novel compared with the previous related studies. This reviewer recommends the manuscript for publication in Nature Communications.

We appreciate the Reviewer's positive feedback on our manuscript.

Reviewer #4 (Remarks to the Author):

We thank the Reviewer for the effort in assessing and improving our paper.